# Distinct and diverse chromatin proteomes of ageing mouse organs reveal protein signatures that correlate with physiological functions

**Giorgio Oliviero†, Sergey Kovalchuk, Adelina Rogowska-Wrzesinska, Veit Schwämmle, Ole N Jensen***

Department of Biochemistry & Molecular Biology and VILLUM Center for Bioanalytical Sciences. University of Southern Denmark, Odense, Denmark

**\*For correspondence:**
jenseno@bmb.sdu.dk

**Present address:** †Systems Biology Ireland, University College Dublin, Dublin, Ireland

**Competing interest:** The authors declare that no competing interests exist.

**Abstract** Temporal molecular changes in ageing mammalian organs are of relevance to disease aetiology because many age-related diseases are linked to changes in the transcriptional and epigenetic machinery that regulate gene expression. We performed quantitative proteome analysis of chromatin-enriched protein extracts to investigate the dynamics of the chromatin proteomes of the mouse brain, heart, lung, kidney, liver, and spleen at 3, 5, 10, and 15 months of age. Each organ exhibited a distinct chromatin proteome and sets of unique proteins. The brain and spleen chromatin proteomes were the most extensive, diverse, and heterogenous among the six organs. The spleen chromatin proteome appeared static during the lifespan, presenting a young phenotype that reflects the permanent alertness state and important role of this organ in physiological defence and immunity. We identified a total of 5928 proteins, including 2472 nuclear or chromatin-associated proteins across the six mouse organs. Up to 3125 proteins were quantified in each organ, demonstrating distinct and organ-specific temporal protein expression timelines and regulation at the post-translational level. Bioinformatics meta-analysis of these chromatin proteomes revealed distinct physiological and ageing-related features for each organ. Our results demonstrate the efficiency of organelle-specific proteomics for in vivo studies of a model organism and consolidate the hypothesis that chromatin-associated proteins are involved in distinct and specific physiological functions in ageing organs.

## Editor's evaluation

The authors have performed an extensive analysis of chromatin enriched proteins as a function of age in mice. Time-course quantitative proteomics reveals the molecular complexity and diversity of mammalian organs and identified aging-related molecular features of chromatin.

## Introduction

Ageing is a natural process resulting in progressive changes of most, if not all, cellular components. Ageing is generally associated with declining biological performance and increased incidence of disease (*Oberdoerffer and Sinclair, 2007*). The gene expression apparatus, comprising the DNA itself, the chromatin environment it is housed in, and the machinery of transcription and translation, is profoundly affected by ageing (*Busuttil et al., 2007*; *Edifizi and Schumacher, 2015*). Models of ageing often display similar phenotypes to those undergoing senescence or genome instability, highlighting how integrated the ageing process is with these phenomena (*Oberdoerffer and Sinclair,*

*2007*). Diseases that mimic or accelerate the ageing process, including Hutchinson–Gilford progeria and Werner syndromes, result in molecular changes in nucleosomes and chromatin (*Arancio et al., 2014*; *Burtner and Kennedy, 2010*; *Feser and Tyler, 2011*).

In eukaryotes, chromatin includes histone molecules that package the DNA, locally controlling access to the underlying genes by facilitating 'open' or 'closed' states associated with transcriptional activation or repression, respectively (*Laugesen and Helin, 2014*). In this way, transcription of individual gene products may be regulated in temporal or location-specific manners.

One of the main routes of proteome expansion is dedicated to enzymes that carry out post-translational modifications (PTMs) of proteins. Enzyme-catalysed PTMs at distinct amino acid residues regulate or modulate protein structure, interactions, and functions (*Santos and Lindner, 2017*).

The mouse, *Mus musculus*, is the most commonly used experimental animal in biomedical research and serves as a model system for studying human health and disease (*Phifer-Rixey and Nachman, 2015*; *Fontana and Partridge, 2015*). Mice have a relatively short lifespan, with one adult mouse month equivalent to approximately three human years (*Sengupta, 2013*; *Dutta and Sengupta, 2016*). This allows for maximum lifespan studies to proceed within the timelines of typical research projects, while environmental factors that affect ageing can be controlled (*Shoji and Miyakawa, 2019*; *Delgado-Morales, 2017*; *Shoji et al., 2016*). Lifespan and health span are mutually influenced by many genes that can either predispose to age-related diseases or slow the ageing process itself (*Murabito et al., 2012*).

We recently applied a middle-down proteomics strategy to demonstrate that mouse chromatin undergoes major changes during ageing, specifically that histone H3.3 replaces H3.1 and that the extent of H3 methylation marks at multiple sites is profoundly altered during ageing (*Tvardovskiy et al., 2017*). We here extend these proteomics studies of mouse chromatin to investigate the protein composition of chromatin in multiple mouse organs during ageing.

We hypothesised that a time-course investigation of the dynamic chromatin proteome could reveal distinct molecular differences of mammalian organs and provide new insights into the regulatory mechanisms in different organs during ageing.

We studied the progressive chromatin protein expression changes in six mouse organs during ageing by quantitative proteomics by mass spectrometry (graphical abstract).

Among almost 6000 proteins identified, organ-specific patterns predominated, with age-responsive subsets identified for each organ. We mapped pathway-level molecular changes specific to individual organ over time.

Our results demonstrate that the ageing process affects each mouse organ in a distinct manner illustrated by the diversity and heterogeneity of the temporal chromatin proteomes of each organ.

## Results

### Isolation of chromatin-associated proteins from mouse cells and organs

We aimed to provide a comprehensive overview of the chromatin-enriched proteome in mouse organs and obtain insights into the molecular processes involved in growth and ageing.

We initially applied a nuclear protein extraction protocol to the mouse embryonic stem cell (mESC) model, which is a 'gold standard' for epigenetics research (*Supplementary file 1*; *Tobin and Kim, 2012*; *Takahashi et al., 2018*). Briefly, mESCs were lysed and cellular compartments were isolated by mechanical disruption followed by a high salt gradient separation to obtain cytosolic, nuclear, chromatin, and histone fractions (*Streubel et al., 2017*; *Herrmann et al., 2017*). These lysates were initially analysed by Western blotting using specific protein markers for each cellular compartment to assess the degree of enrichment of chromatin-associated proteins (*Figure 1A*).

We performed quantitative proteomics by triplicate high-mass-accuracy mass spectrometry analysis of the mESC proteome and the mESC chromatin proteome to assess the enrichment of chromatin-associated proteins (*Figure 1B and C*). We present the detected chromatin enzyme/protein complex components in *Table 1*. We annotated all nuclear or chromatin-associated proteins using available database resources (*Figure 1—figure supplement 1*; *Medvedeva et al., 2015*; *van Mierlo et al., 2019*; *Christoforou et al., 2016*; *Xu et al., 2016*).

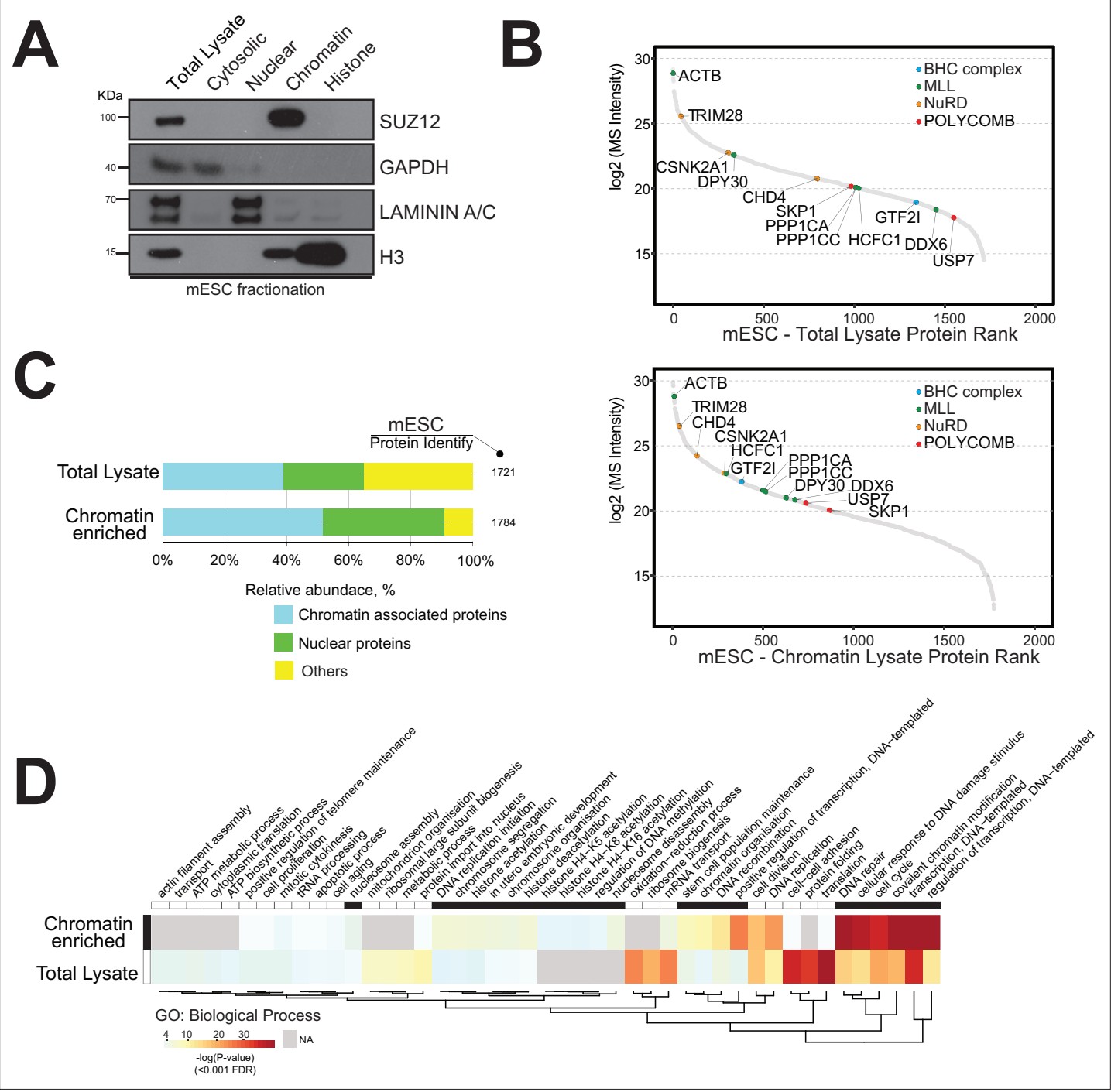

**Figure 1.** Strategy for the enrichment of chromatin-associated proteins in mouse embryonic stem cells. (**A**) Western blot analysis of mouse embryonic cell fractionation. The cytoplasmic, nuclei, chromatin, and histone compartments are probed (respectively) by GAPDH, Laminin A/C, SUZ12, and H3 antibodies. (**B**) Dynamic range plot of mouse embryonic cell fractionation. Epigenetic enzyme subunits associated with chromatin remodelling complexes are listed and sorted by their abundance (Log2 MS intensity) for the two datasets: chromatin proteome lysate and whole-cell lysate. (**C**) Evaluation of the enrichment of chromatin-associated proteins in mouse embryonic cell fractions. The proportion of chromatin-associated and nuclear proteins in each organ is shown. Blue, chromatin-associated protein; green, nuclear proteins; yellow, protein associated with other cellular components. The relative abundances were determined based on the total ion current. The obtained quantitative results were used to calculate the relative abundances of distinct chromatin-associated and nuclear proteins corresponding to each cell compartment, where the sum of all total ion current intensities was considered as 100%. (**D**) Hierarchical clustering heatmap of mouse embryonic cell fractionation. Chromatin fractionation and total lysate proteomes are compared. Significantly enriched Gene Ontology (GO) term biological processes (BP) associated with nuclear and chromatin environments are enriched in the chromatin fraction lysate.

*Figure 1 continued on next page*

*Figure 1 continued*

The online version of this article includes the following source data and figure supplement(s) for figure 1:

**Source data 1.** Western blot data.

**Figure supplement 1.** Comparison of annotated chromatin-associated proteins from recently deposited proteomics data libraries of chromatin studies.

**Figure supplement 2.** Evaluation of the enrichment of chromatin-associated proteins in mouse organs.

**Figure supplement 2—source data 1.** Western blot results (original scans).

The dynamic range plot was used to assess the measurements of protein expression across these proteomes. The mESC chromatin fraction was indeed highly enriched for chromatin-associated proteins compared to the whole-cell lysate (*Figure 1B and C*).

We detected major chromatin-associated protein complexes, including the polycomb repressive complex 2 (PRC2), nucleosome remodelling and deacetylase (NuRD), BRAF-HDAC complex (BHC), and mixed-lineage leukaemia (MLL) complex (*Figure 1B*; *Medvedeva et al., 2015*; *van Mierlo et al., 2019*; *Xu et al., 2016*).

In general, we observed an overall increase in the proportion of proteins classified as either 'nuclear' (20%) or 'chromatin-associated' (35%) within the mESC chromatin sample (*Figure 1*).

Gene Ontology (GO) analysis showed a distinct enrichment of proteins associated with 'DNA-protein binding', 'histone binding', and 'chromatin and nucleosome organisation' within the chromatin sample (*Figure 1D*). The mESC total lysate sample mainly contained proteins involved in 'translational protein', 'cell-cell structure organisation', and 'ribosomal and ATP processes' (*Figure 1D*).

We next demonstrated that the chromatin enrichment protocol used for mESCs is applicable to mouse organs. We first extracted chromatin proteins from mouse brain and assessed chromatin enrichment by Western blotting using specific protein markers (*Figure 1—figure supplement 2*). We observed enrichment of the chromatin marker histone H3 and reduced level of the cytosolic marker GAPDH expression within the 'chromatin fraction' lysate.

Next, we performed quantitative mass spectrometry profiling of the mouse brain proteome and the mouse brain chromatin-enriched proteome. The dynamic range plots demonstrated that

**Table 1.** Number of chromatin enzymes detected in mouse embryonic stem cell (mESC).

| Chromatin remodelling complex | Whole-cell lysate | Chromatin-enriched lysate |
| --- | --- | --- |
| BHC | 2 | 6 |
| CCR4-NOT | 2 | 6 |
| ING | 1 | 15 |
| INO80 | 2 | 9 |
| MEDIATOR | 8 | 20 |
| MLL | 15 | 56 |
| NUA4HAT | 17 | 8 |
| NURD | 2 | 35 |
| NURF | 10 | 2 |
| ORC | 1 | 30 |
| POLYCOMB | 7 | 30 |
| READER | 8 | 33 |
| SAGA | 8 | 8 |
| SET1 | 3 | 6 |
| SIN3A | 1 | 26 |
| SWI/SNF | 5 | 43 |
| Other chromatin enzymes | 10 | 80 |

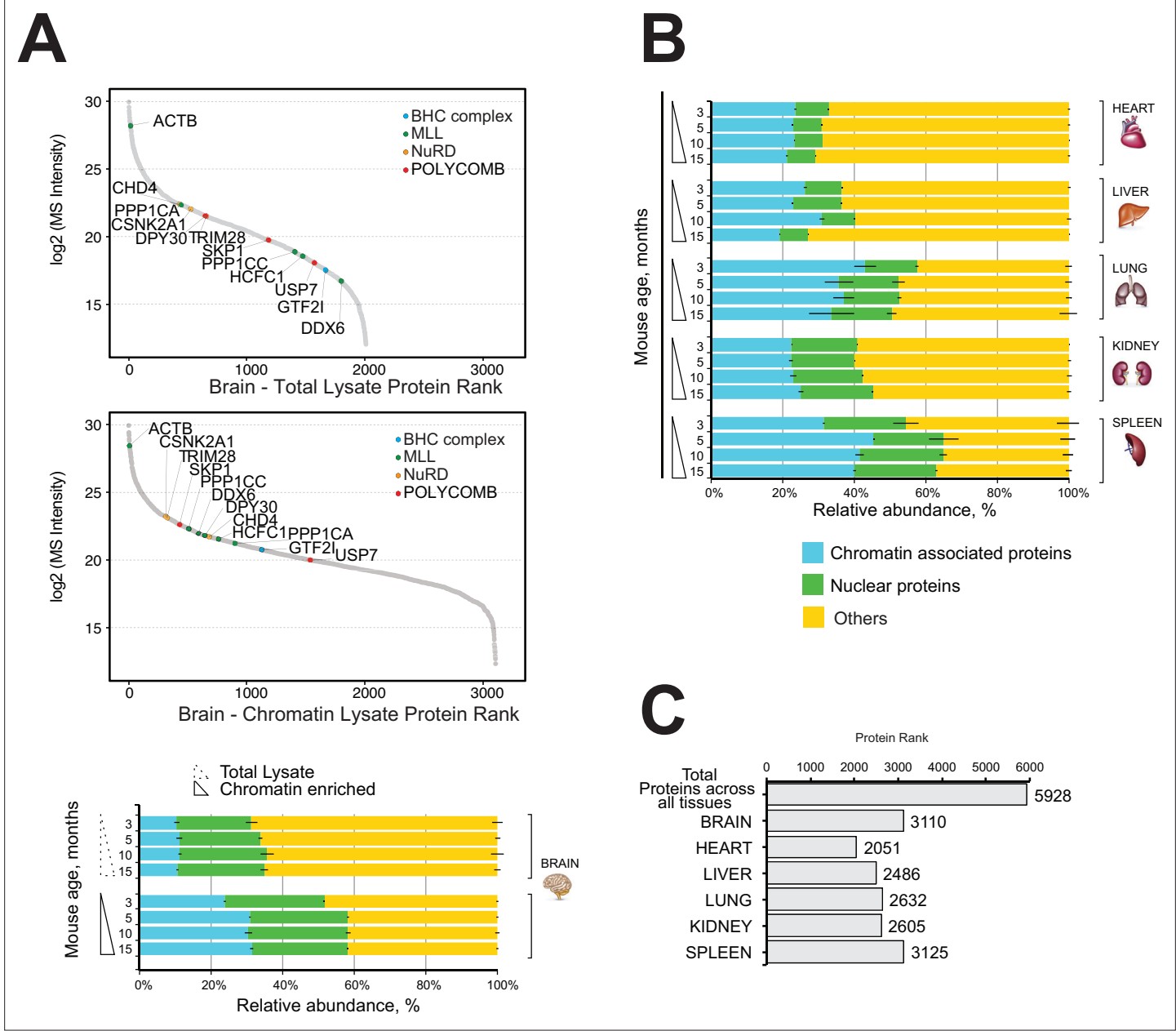

**Figure 2.** Comparative and quantitative proteomics of chromatin proteins in mouse organs. (**A**) High-resolution LC-MS/MS brain proteomics during mouse lifespan. Dynamic range plot of the brain tissue fractionation obtained from a mouse. Chromatin lysate and whole total lysate proteomes are compared. Epigenetic subunits associated with chromatin remodelling complexes are listed and sorted by specific abundance (Log2 MS Intensity) in both proteomics datasets. Blue, chromatin-associated protein; green, nuclear proteins; yellow, protein associated with other cellular components. The continuous line box indicates the total lysate proteome, and the dashed line box indicates the chromatin fractions proteome. (**B**) High-resolution LC-MS/MS chromatin proteome harvested from 3-, 5-, 10-, and 15-month-old mice. Evaluation of the amount of chromatin-associated proteins and nuclear protein among different organs (heart, liver, lung, kidney, and spleen). The relative abundances were quantified based on the total ion current. The obtained quantitative results were used to calculate the relative abundances of distinct chromatin-associated and nuclear proteins in each organ, where the sum of all total ion current intensities was considered as 100%. Blue, chromatin-associated protein; green, nuclear proteins; yellow, protein associated with other cellular components. The continuous line box indicates total lysate proteome, and the dashed line box indicates the chromatin fractions proteome. (**C**) Histogram showing the number of proteins identified across six mouse organs over time.

the major chromatin-binding protein complexes were enriched in the brain chromatin sample, including MLL, NuRD, Polycomb, and BHC complexes (*Figure 2A*, *Supplementary file 2*, Table S2).

The mouse brain chromatin-enriched fraction contained more than 30% chromatin-associated proteins, up from ~8% in the total brain lysate (*Figure 2A*, bottom). This was accompanied by a large increase in the content of 'nuclear proteins' (~20%) (*Figure 2A*).

We conclude that our chromatin proteome fraction of mouse brain tissue was highly enriched in chromatin-associated and nuclear proteins compared to the whole brain lysate (*Figure 2A*).

## Quantitative chromatin proteomics of ageing mouse organs

We performed quantitative chromatin proteomics of six mouse organs to investigate the in vivo dynamics of chromatin during ageing. We isolated chromatin-associated proteins from mouse brain, heart, liver, kidney, lung, and spleen at time points 3, 5, 10, and 15 months representing the 'mature adult mouse lifespan', from the early adult stage (3 months), middle-aged adult (5–10 months), and mature adult (15 months) (graphical abstract) (*Dutta and Sengupta, 2016*; *Shoji and Miyakawa, 2019*; *Delgado-Morales, 2017*; *Shoji et al., 2016*).

We excluded mice older than 18 months to minimise any age-related changes that might be due to social behaviour, physical characteristics of motor function, and locomotor activity.

Proteins were identified and quantified by high-mass-accuracy LC-MS/MS by hybrid quadrupole-orbitrap technology using a peptide intensity-based (label-free) protein quantification strategy (see Materials and methods, *Supplementary file 3*, Table S3).

The proportion of nuclear proteins or chromatin-associated proteins ranged from 30% to 60% of all detected proteins, and it was similar across all time points for each organ (*Figure 2B*).

Subsequent data analysis included all identified proteins of each organ and all time points to avoid loss of essential information and achieve a more detailed characterisation of the chromatin proteomes of mouse organs.

We identified a total of 5928 proteins in the chromatin-enriched protein samples across all six mouse organs over time (*Figure 2C*). Most proteins were identified in the chromatin fractions of mouse brain (3110) and spleen (3125), whereas the lowest number of proteins were identified in the chromatin fraction of mouse heart organ (2051) (*Figure 2C*). The lung and spleen samples were highly enriched in nuclear proteins and chromatin-associated proteins (55–65%), whereas the chromatin-enriched heart sample contained 30–35% nuclear/chromatin-associated proteins.

The very different morphology and cell-type compositions of the mouse organs likely influence the efficiency of the chromatin protein extraction protocol and thereby the detected proteome compositions. Cell-type and cell cycle-specific transcriptional activities likely explain some of the observed variation in proteome composition.

Nevertheless, the fraction of proteins classified as either 'nuclear' or 'chromatin-associated' was similar among all mouse organs and time points, demonstrating the high reproducibly and reliability of the experimental approach with a coefficient of variation estimated less than 10% among all the samples (*Figure 2B*).

Overall, we identified a total of 4581 different chromatin-associated proteins across all organs, including 2717 different nuclear proteins (*Table 2*).

**Table 2.** Number of proteins detected across six mouse organs over four time points and their annotated subcellular location.

Detected number of unique proteins across all organs are shown for each subcellular location.

**Chromatin proteomes during the mouse lifespan**

| Organ | Brain | Heart | Liver | Kidney | Lung | Spleen | Unique proteins across all organs |
|-------|-------|-------|-------|--------|------|--------|------------------------------------|
| Chromatin | 936 | 457 | 648 | 714 | 713 | 1113 | 1542 |
| Nuclear | 615 | 323 | 355 | 426 | 442 | 556 | 930 |
| Other | 1559 | 1272 | 1483 | 1465 | 1477 | 1457 | 3456 |

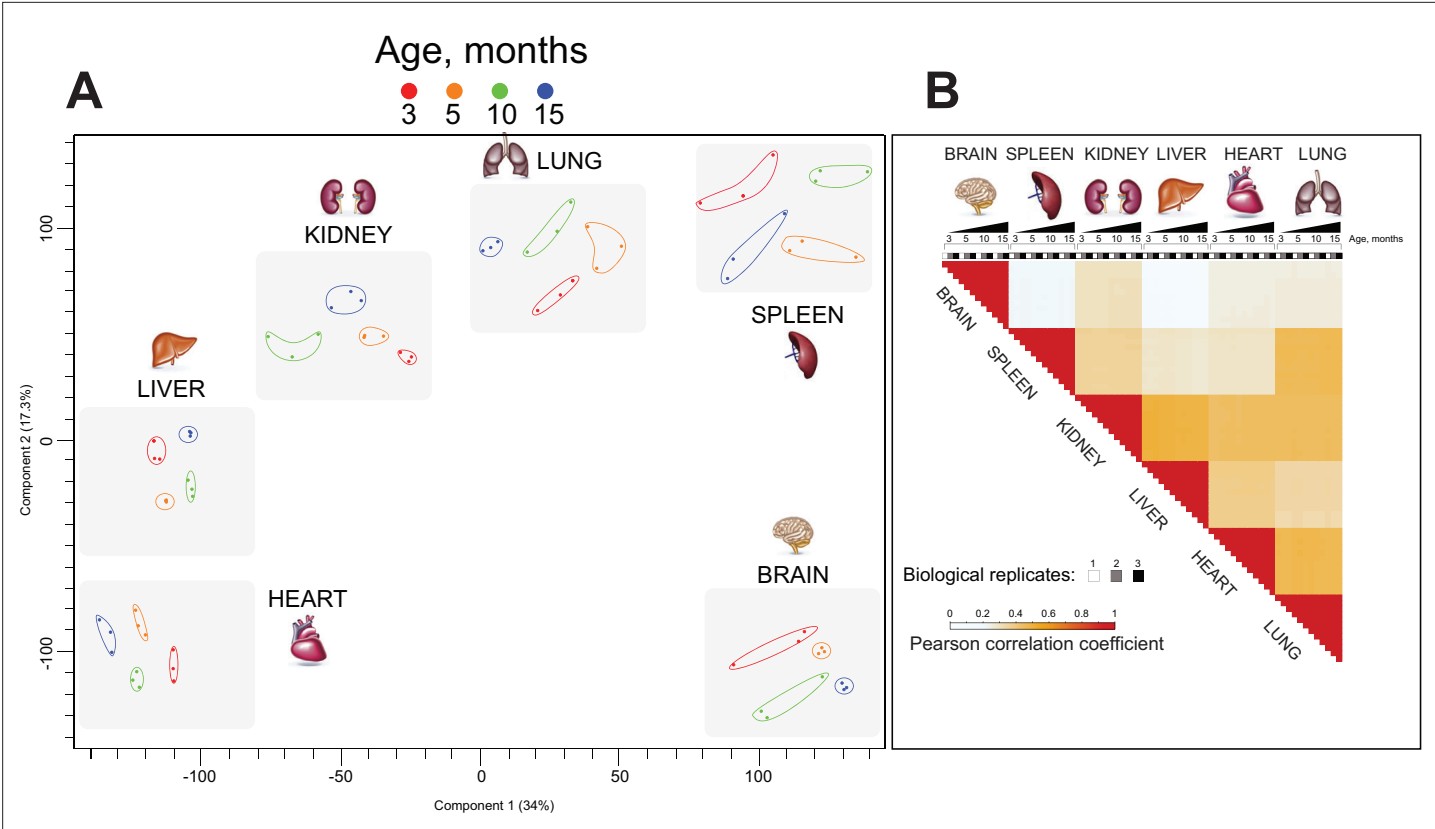

**Figure 3.** Chromatin proteomics by high mass resolution mass spectrometry demonstrated distinct organ ageing profiles. (**A**) Principal component analysis (PCA) of the proteomics datasets. Each data point represents a single replicate (n = 3). Colour subgroups represent each mouse lifespan point, being 3, 5, 10, and 15 months, respectively. The grey-coloured square behind each replicate highlights a distinct separation between each organ during the mouse lifespan. (**B**) Pearson correlation coefficient showing the relationship between the different enriched chromatin proteome organs and ageing. The positive correlation coefficient is displayed in red, and reduced values are shown in bright blue and white.

The online version of this article includes the following figure supplement(s) for figure 3:

**Figure supplement 1.** Evaluation of robustness and reproducibility of quantitative proteomics LC-MS/MS strategy.

## Proteomics reveals organ-specific protein profiles during ageing

We assessed the entire mouse organ chromatin-enriched proteome dataset using principal component analysis (PCA) (*Figure 3A*). The time points (age of mouse at organ harvest) were well separated from one another for each organ, such that the replicates for each time point were more closely clustered to one another than to replicates of other time points. More striking was the observation that the origin of organ was the fundamental discriminating factor for the overall clustering of samples: each organ formed its distinct cluster made up of subclusters comprising the different ages of the organ samples (*Figure 3A*).

Pearson correlation analysis (*Figure 3—figure supplement 1*) demonstrated the reproducibility of biological replicates and confirms the robustness of our biochemical and proteomics methodology. To further characterise similarities between ageing organ and proteome expression profiles, we performed clustering of Pearson correlation coefficients (*Figure 3B*). This showed the consistency and reproducibility of analysis of three biological replicates at all time points and revealed the distinct organ proteome profiles. The brain-derived proteome exhibited poor correlation to all other organs. Kidney, liver, heart, and lung samples exhibited protein expression profiles, which were slightly positively correlated. Spleen displayed a slightly positive correlation with kidney and lung and poor correlation to the brain, liver, and heart.

Overall, our PCA and Pearson correlation analyses demonstrate that each mouse tissue exhibits a distinct and ageing-related chromatin proteome profile (*Figure 3*).

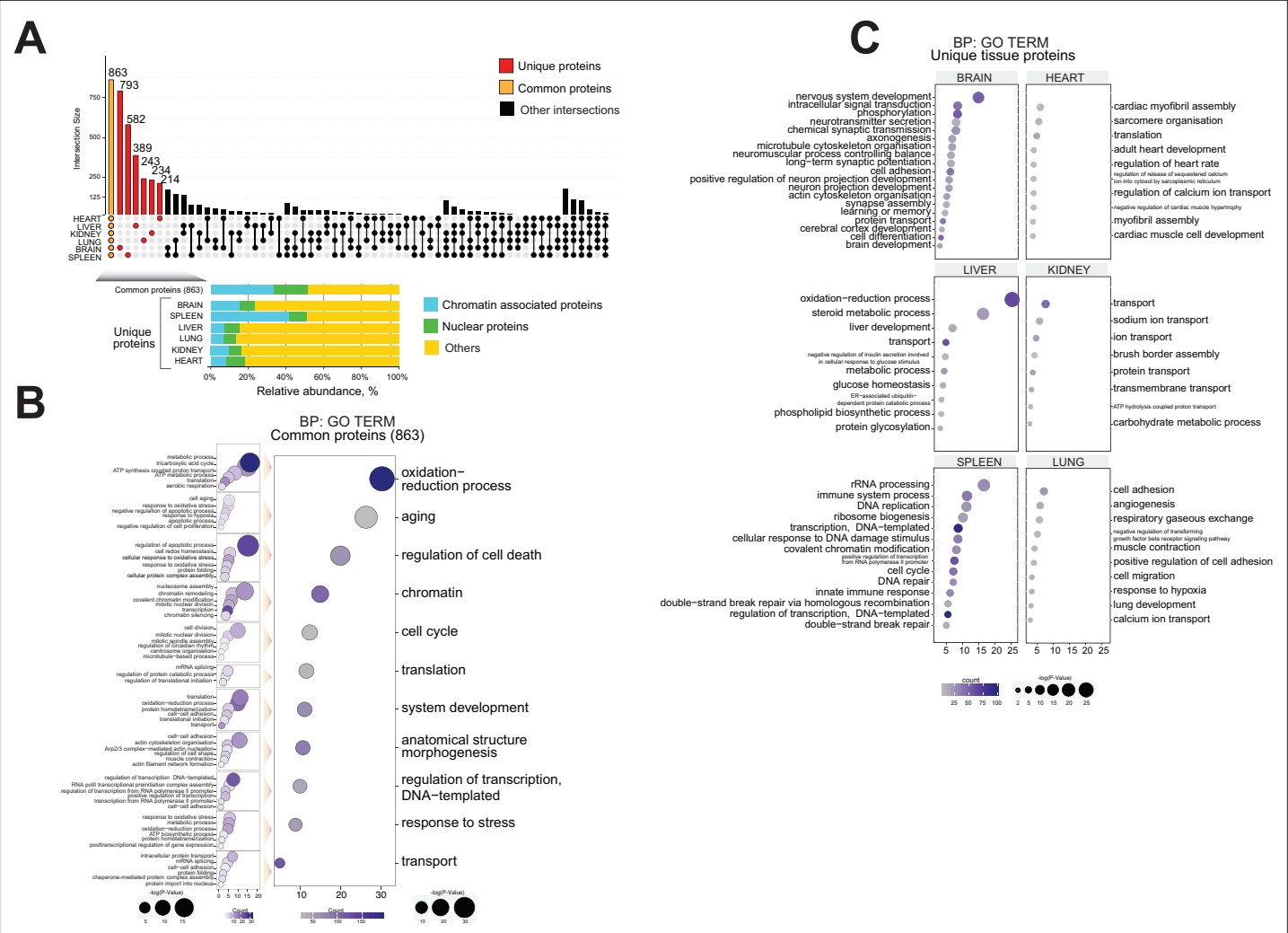

**Figure 4.** Functional proteome analysis of common and distinct organ ageing profiles during the mouse lifespan. (**A**) Overlap of proteome data sets across six mouse organs using an UpSet plot. The number of common (orange) and unique (red) organ-specific proteins detected is shown, while various inter-organ combinations are displayed in black. In the bottom of the panel, the proportion of chromatin-associated proteins and nuclear proteins present for the 'core' proteome and the unique organ-specific profile is shown. The relative abundances were quantified based on the total ion current. The obtained quantitative results were used to calculate the relative abundances of distinct chromatin-associated and nuclear proteins in each organ, where the sum of all total ion current intensities was considered as 100%. Blue, chromatin-associated protein; green, nuclear proteins; yellow, protein associated with other cellular components. (**B**) Gene Ontology (GO) analysis (biological processes) of the core proteome (863) was performed by DAVID GO analysis. The right panel indicates the most significant GO term categories. The left panel shows multi sub-annotations corresponding to each category. Dot size represents the logarithm of the p-value assigned to the detected category, while dot colour represents the number of proteins in the pathway. (**C**) GO analysis (biological processes) of the unique proteins present in each organ proteome. The dot size represents the significant p-value assigned to the detected category, while the dot colour represents the number of proteins corresponding to the source pathway.

## Quantitative proteomics defines biological changes in the ageing process

To define ageing signatures across mouse lifespan in each organ, we performed a comprehensive functional analysis of the proteomics datasets using an UpSetR plot to investigate ageing markers (*Figure 4A*; *Conway et al., 2017*).

Briefly, comparative organ proteome analysis was performed to estimate the number of shared proteins and to determine the degree of similarity between datasets and, subsequently, to identify unique organ-specific feature. Then, the datasets were investigated by GO analysis to explore the relationship between chromatin protein composition and ageing.

**Table 3.** 'Core' proteins shared in all six mouse organs over time sorted to their cell compartments.

| | Chromatin proteome during the mouse lifespan | | | | | | |
| | Shared proteins | Unique proteins | | | | | |
| | Core | Brain | Heart | Liver | Kidney | Lung | Spleen |
|---|---|---|---|---|---|---|---|
| Chromatin | 289 | 119 | 18 | 27 | 22 | 17 | 242 |
| Nuclear | 157 | 69 | 21 | 33 | 16 | 15 | 54 |
| Other | 417 | 605 | 175 | 329 | 196 | 211 | 286 |
| | | | | | | | |
| Total | 863 | 793 | 214 | 389 | 234 | 243 | 582 |

A 'core' proteome of 863 proteins was identified across all six organs during mouse lifespan across four time points, including 289 shared chromatin-binding proteins and 157 shared nuclear proteins (*Figure 4A*, *Table 3*).

The core chromatin proteome contained proteins shared across all tissue during the mouse lifespan. These proteins were associated with the major transcriptional/epigenetic chromatin complexes such as BHC (GTF2I), MLL (ACTB, RUVBL2, DPY30, WDR82, PPP1CA, SEPT9, PPP1CC, PPP1CB, SNX2, LPP, DDX6), NuA4HAT (RUVBL1), NuRD (CHD4, TRIM28, CSNK2A1), NURF (SMARCA5), PcG (RBBP4, RBBP7), SAGA (FGG), SIN3A (MECP2, SFPQ, PA2G4), and SWI/SNF (SMARCC2, YWHAB, H2AFY, RAC1, EIF4B) (*Medvedeva et al., 2015*; *van Mierlo et al., 2019*; *Xu et al., 2016*; *Spruijt et al., 2016*; *Varier et al., 2016*). Many of these proteins are expressed during cell fate commitment (*Medvedeva et al., 2015*; *Signolet and Hendrich, 2015*).

For each of the six organs, we detected from 214 to 793 proteins that were unique to that organ, that is, proteins not detected in other organs (*Figure 4A*, *Table 3*).

The mouse brain chromatin proteome contained 793 unique proteins, constituting the largest set of unique proteins among the six organs (*Figure 4A*). Approximately 24% of these proteins (188) were classified as chromatin-associated proteins or nuclear proteins (*Figure 4A*, *Table 3*). They included protein and histone methyltransferase enzymes (HNMT, SAP30L, CARM1, SETD7, SUV39H2) and histone deacetylase enzymes (CHD5, IRF2BP1, MAPK10, MEF2D, MACROD2) that are mainly involved in transcriptional gene silencing (*Medvedeva et al., 2015*; *van Mierlo et al., 2019*; *Xu et al., 2016*; *Hyun et al., 2017*).

The spleen chromatin preparation contained 582 unique proteins, ~50% of which are chromatin-associated proteins or nuclear proteins, strongly suggesting a distinctive chromatin proteome profile for this organ (*Figure 4A*, *Table 3*).

The spleen is an organ with the innate capacity to regenerate (*Holdsworth, 1991*). It acts as a filter for blood and controls the blood-borne immune response (*Tan and Watanabe, 2018*). We detected several epigenetic complexes such as BHC, ING, MLL, NuRD, ORC, PCG, SIN3A, SAGA, and SWI/SNF. Also, we detected chromatin 'reader' enzymes not yet assigned to a specific chromatin remodelling complex (KDM3B, KDM2A, PHF23, CHD1L, UHRF2, MORC3, BRD9, ZMYND11, BAZ1A). These proteins recognise single post-translational histone marks or combinations of histone marks and histone variants to direct a particular transcriptional outcome (*Medvedeva et al., 2015*; *van Mierlo et al., 2019*; *Xu et al., 2016*; *Hyun et al., 2017*).

This result is in line with our above observations, suggesting that spleens were highly enriched in epigenetic markers and therefore are a good model to study chromatin remodelling complexes. Ageing effects are difficult to distinguish in spleen tissue.

The relatively large numbers of unique proteins of each organ likely reflect the inherent features of the individual organs, the diversity of cell types, and physiology (*Figure 4A*).

GO analysis of the 'core' 863 proteins revealed a large number of 'chromatin-associated' or 'nuclear' proteins. The GO output was enriched for categories related to mouse ageing, such as 'oxidation-reduction process', 'ageing', 'regulator of cell cycle', and 'stress response' (*Figure 4B*; *Go and Jones, 2017*; *Haigis and Yankner, 2010*; *Epel and Lithgow, 2014*; *Postnikoff and Harkness, 2012*; *Chandler and Peters, 2013*).

Subsequently, each category of the core proteome was further broken down into its constituent parts to create a map of the shared ageing-related molecular network of the six mouse organs (*Figure 4B*). Our aim was to identify novel biological features and associate them with ageing-related pathways and annotations.

GO classification of the unique proteins present in each organ proteome suggested distinctive molecular signs of ageing in each organ (*Figure 4C*). We observed distinctive organ-specific categories, 'age classes and age development', and categories that reflected their organ source (*Figure 4C*). For instance, unique GO term categories were associated with each organ, such as 'nervous system development' and 'chemical synaptic transmission' related to the brain; 'cardiac myofibril assembly' and 'adult heart development' were attributed to the heart; 'steroid metabolic process' and 'liver development' were distinctive to the liver; 'Transport' and 'sodium ion transport' categories related to the kidney; and 'angiogenesis' and 'respiratory gaseous exchange' were present in the lung.

These results confirm that many chromatin proteins found in individual organs likely confer organ-specific functions (*Figure 4C*).

Taken together, our proteomics analysis showed a robust enrichment of chromatin-associated proteins in mouse organs as confirmed by GO term analysis. We reported a significant enrichment of age-related proteome features, including a large class of protein annotations associated with the core chromatin environment present in all organs.

## Distinct organ ageing profiles are defined by unique protein expression patterns

We hypothesised that different mammalian organs have ageing-dependent and distinct expression profiles of characteristic chromatin-associated proteins.

We employed a temporal analysis of the overall dataset which included chromatin-associated proteins, nuclear proteins, and unassigned proteins to uncover the common features that may contribute to the chromatin environment during ageing.

We used the rank products test to identify the proteins that exhibited significant abundance changes in one direction during the ageing of the organ or linearly throughout the lifespan of the animal (*Figure 5—figure supplement 1A and C*; *Koziol, 2010*; *Breitling et al., 2004*). We call these 'differentially regulated proteins' (*Figure 5*, *Figure 5—figure supplement 2A*).

We then retrieved those unique regulated proteins that were specifically detected in only one organ, two organs, or three organs (UpSet plot) (*Figure 5A*). The majority of uniquely regulated proteins were indeed specific to one organ.

We subjected the organ-specific uniquely regulated proteins to hierarchical clustering based on their expression changes and depicted them as heatmaps for each organ (three replicates per time point) (*Figure 5B*). This allowed us to compare how the relative expression of unique chromatin-associated proteins changed over time in each organ, from early to late time points (3–15 months) (*Figure 5B*, *Supplementary file 4*, Table S4).

The hierarchical clustering shows that the number of up- and downregulated proteins is similar in each organ. Further, assigned cell compartments of the 'unique differentially regulated proteins' are shown as sidebars (*Figure 5B*, *Supplementary file 4*, Table S4). A large number of non-annotated 'unique differentially regulated proteins' show a similar quantitative behaviour across all organs. We hypothesised that these proteins may represent a useful list of candidates that may regulate gene expression by being transiently recruited to chromatin at distinct time points during ageing.

Brain, heart, and kidney showed mainly gradual changes of protein levels over time. Lung and liver show a more dramatic change of protein expression between 5 and 10 months of age (*Figure 5B*, *Figure 5—figure supplement 2B*). We detected a few significant protein abundance changes in the spleen. The spleen fraction was highly enriched in chromatin-associated and nuclear proteins (~60%) and contained many unique proteins (*Figures 3 and 4*). Thus, the spleen seems to continuously exhibit a young phenotype that may be due to the constitutively active role of the spleen in maintaining immune functions, red blood cell turnover, and microbial defence (*Turner and Mabbott, 2017*). The spleen contains multiple cell populations capable of supporting immune responses, which may indicate the presence of self-renewal cell types that are 'age-less' (*Holdsworth, 1991*).

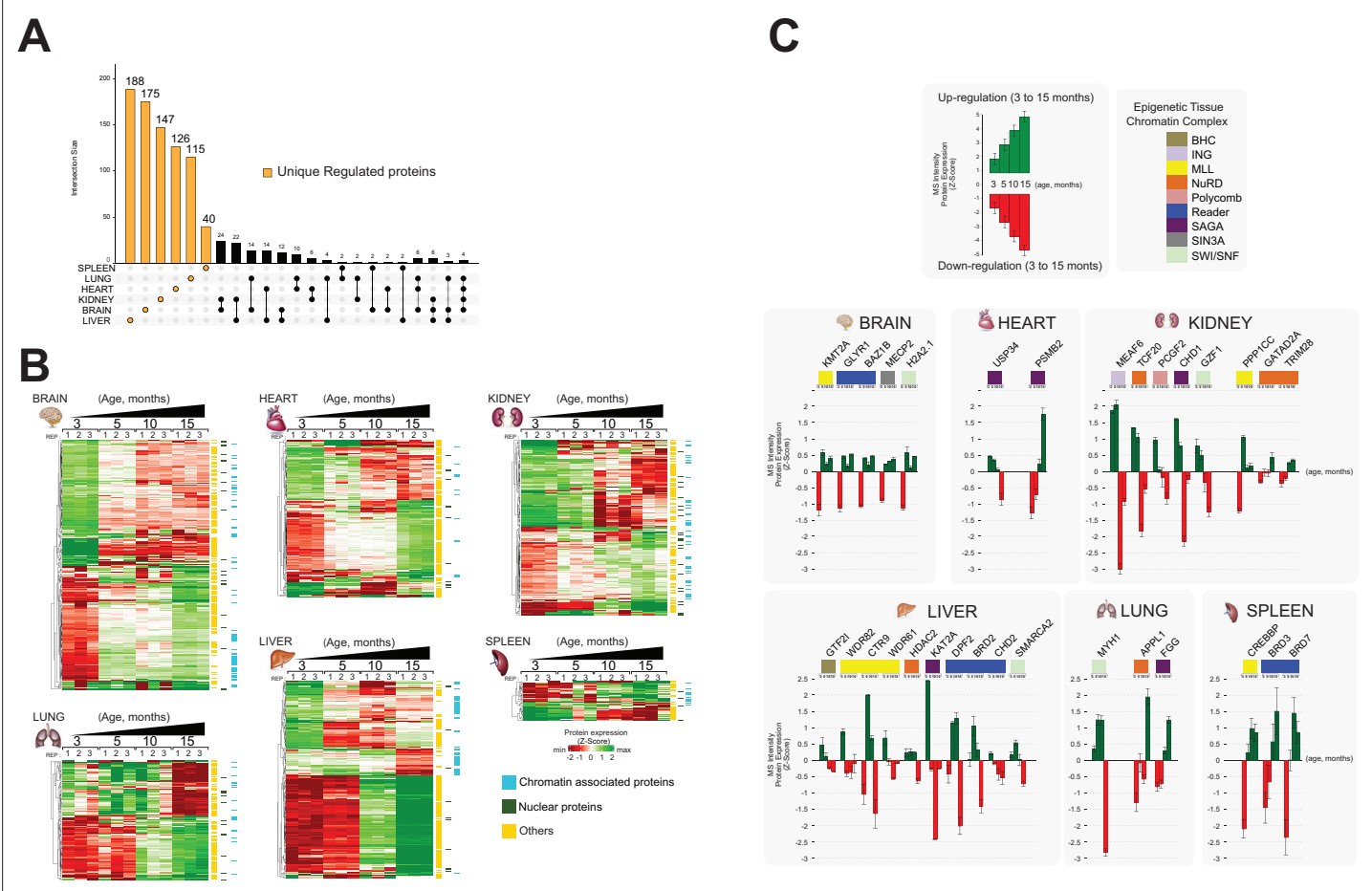

**Figure 5.** Identification of unique protein abundance changes in organ proteome profiles during ageing. (**A**) UpSet plot measurement of the amount of nuclear and relative chromatin-associated proteins differentially expressed during the mouse lifespan across all organs. Unique chromatin-associated proteins related to their organ sources are highlighted by an orange bar. The multiple intersection nodes highlighted in black display shared proteins between organs. (**B**) Hierarchical clustering heatmap of the 'unique differentially expressed protein' related to each organ among the early (3 months) and the late (15 months) ageing stages. Green and red represent increased and decreased expression (respectively) during the mouse lifespan. Blue, chromatin-associated protein; green, nuclear proteins; yellow, protein associated with other cellular components. (**C**) Extrapolation of the quantitative expression profile of epigenetic subunits associated with chromatin remodelling complexes. Green and red represent increased and decreased expression (respectively) during the mouse lifespan.

The online version of this article includes the following figure supplement(s) for figure 5:

**Figure supplement 1.** Evaluation of protein expression changes in mouse organs during ageing.

**Figure supplement 2.** Distinct organ ageing profiles are defined by unique differential protein expression.

In summary, we identified a large number of unique differentially regulated proteins in the chromatin-enriched proteomes of mouse organs. The abundance of these proteins changes dramatically during ageing from month 3 to month 15, across all organs, except for spleen.

Next, we explored the potential functional links between chromatin proteome dynamics and ageing. We investigated all chromatin-associated proteins that are part of known well-defined chromatin complexes that were identified among the 'unique differentially regulated proteins'.

The majority of chromatin-modifying enzymes belonging to a given multiprotein complex exhibited similar expression profiles over time within a specific organ (***Figure 5C***). For instance, the MLL subunits WDR82, CTR9, and WDR61 were downregulated in the liver during ageing. Components of the same chromatin-modifying complexes were detected in several organs, albeit not by the same subunits and with opposite temporal expression profiles. NuRD subunit HDAC2 was downregulated in the liver, whereas NuRD subunits GATA2AD and TRIM28 were upregulated in the kidney.

This is consistent with the highly dynamic nature and spatio-temporal regulation of chromatin remodelling complexes. Some protein subunits are only present in a complex at distinct time points to provide a unique function or feature (*Oliviero et al., 2016*).

A series of 'Reader' enzymes were upregulated in the brain (GLYR1, BAZ1B) and spleen (BRD3, BRD7), whereas other 'Reader' enzymes were downregulated in the liver (DPF2, BRD2, CHD2) (*Figure 4C*).

Next, we queried the 'Human Ageing Genomic Resources' and 'GenAge machine learning databank' using our complete list of 'unique regulated proteins' that are not yet assigned to chromatin or nuclear environment to demonstrate the ability of our mouse organ proteomics approach to detect known human ageing biomarkers (*Figure 5—figure supplement 2C*; *Tacutu et al., 2013*; *Kerepesi et al., 2018*).

We identified a series of human protein biomarker candidates for ageing. The brain protein IREB2 is associated with Alzheimer's disease, whereas the brain protein MAOB is associated with both Alzheimer's and Parkinson's diseases. The heart proteins ADD3, PTGIS, and COL1A2 are candidates for hypertension and myocardial infarction. The liver proteins INSR, PTPN1, and ENPP1 are associated with diabetes mellitus type 2 and obesity. Lung protein CYP2E1 is related to lung adenocarcinoma, and MMP9 is associated with lung neoplasms. In the spleen, KLK1 protein is a biomarker candidate for hypertension.

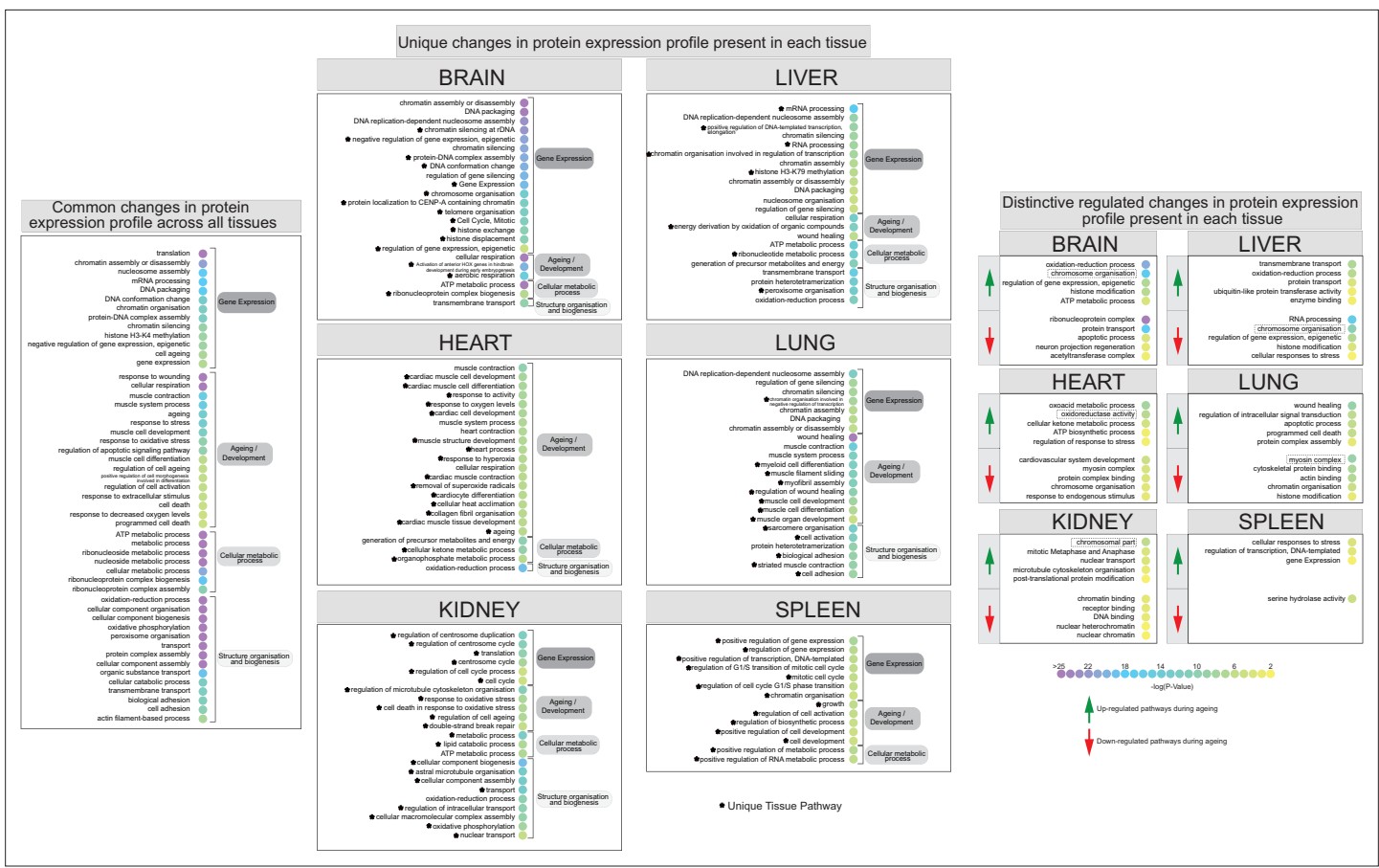

**Figure 6.** Identification of differentially regulated organ-specific profiles that map to ageing, epigenetic processes, and other biochemical processes. Dot plot of functional annotation Gene Ontology (GO) analysis of the biological processes (BP) showing the enrichment pathway terms among ageing. GO categories are sorted by four group labels: gene expression, ageing/development, cellular metabolic process, and structure organisation and biogenesis. The left panel indicates the significant top 30 annotation categories shared between all organs and changing during the mouse lifespan. The centre panel shows the distinctive ageing pathway related to their organ sources. The right panel shows the enrichment pathway terms that changed during ageing among all organs. Red and green highlights represent the protein downregulation and upregulation between early (3 months) and late (15 months) ageing stages. The dot colour represents the significant p-value of the pathway.

## Functional analysis of chromatin-enriched proteomes of ageing mouse organs

Next, we applied GO analysis to characterise all 'unique differentially regulated proteins' detected in each organ (*Figure 6*).

We listed the overall common pathways and processes that were found by quantitative chromatin proteomics to be differentially regulated during ageing across all the organs. We sorted the annotated features by their relative GO term category and separated them by their main family source (*Figure 6*, left panel).

During adult mouse lifespan, we observed several ageing stress response pathways and GO categories associated with the regulation of chromatin architecture that may affect cellular structures and morphology.

By listing every single category, we could describe the biological profile and pathways affected by the age-related protein expression responses present in all organs (*Figure 6*).

Subsequently, the differentially regulated proteins were sorted by their organ source and subject to further GO term analysis to distinguish unique organ-related processes from those pathways associated with ageing (*Figure 6*, centre panel).

We report a high proportion of uniquely annotated categories for each mouse organ. For instance, the highest unique changes observed in the brain were relative to 'gene expression' and 'ageing/development'; in the heart and kidney, significant changes were observed relative to 'structure organisation and biogenesis'; the liver showed changes across the 'gene expression' and 'structure organisation and biogenesis' categories; the lung showed the highest unique changes in the 'ageing/development' category, and relative changes in the spleen were detected at the 'gene expression' level.

These results are in line with our above observations, suggesting a unique ageing response from each organ as evidenced by distinct dynamic changes of chromatin-associated proteins.

We further interrogated the list of unique annotated organ categories to highlight the distinct and significant temporal pathway profiles among the up- and downregulated proteins in each organ to reveal the most distinctive regulated features (*Figure 6*, right panel).

In the mouse brain tissue, proteins involved in pathways such as 'chromosome organisation' and 'histone modification' were strongly upregulated, while those involved in the regulation of 'neuron projection regeneration' were downregulated. In mouse heart tissue, upregulated protein pathways included 'oxidoreductase activity' and 'regulation of response to stress', while downregulated pathways included 'cardiovascular system development' and 'chromosome organisation'. In the kidneys, proteins involved in pathways related to changes in chromatin conformation were both going up and going down during the mouse lifespan. In the liver, we observed that downregulated proteins were associated with 'chromosome organisation' and 'histone modification', while 'oxidation-reduction process' pathways were upregulated. In the lungs, we noticed that proteins associated with 'apoptotic process' and 'programmed cell death' were upregulated, while pathways related to 'muscle organisation and reassemble' were downregulated.

Similarly, several different processes were altered in the other organs, except for the spleen that did not show many significant changes during ageing. In line with previous data, little pathway-level changes were observed in the spleen, especially in the downregulated proteins, possibly indicating the important role of this organ in removing old red blood cells and microbes which seem, from our data, to not be affected by ageing. For this reason, the spleen was not considered for further analysis.

Overall, using GO term analysis we dissected the biological features of the chromatin proteomes of organs in the context of mouse lifespan. By breaking down common and unique regulated functional categories, we surveyed ageing-related pathways and improved gene-annotation enrichment analyses.

In conclusion, we identified and measured distinct and extensive protein abundance changes during ageing, specifically in early to mature adult mouse lifespan. A large number of differentially expressed proteins are unique for each organ as defined by specific GO term categories. This demonstrated that the ageing process affects each mouse organ differently.

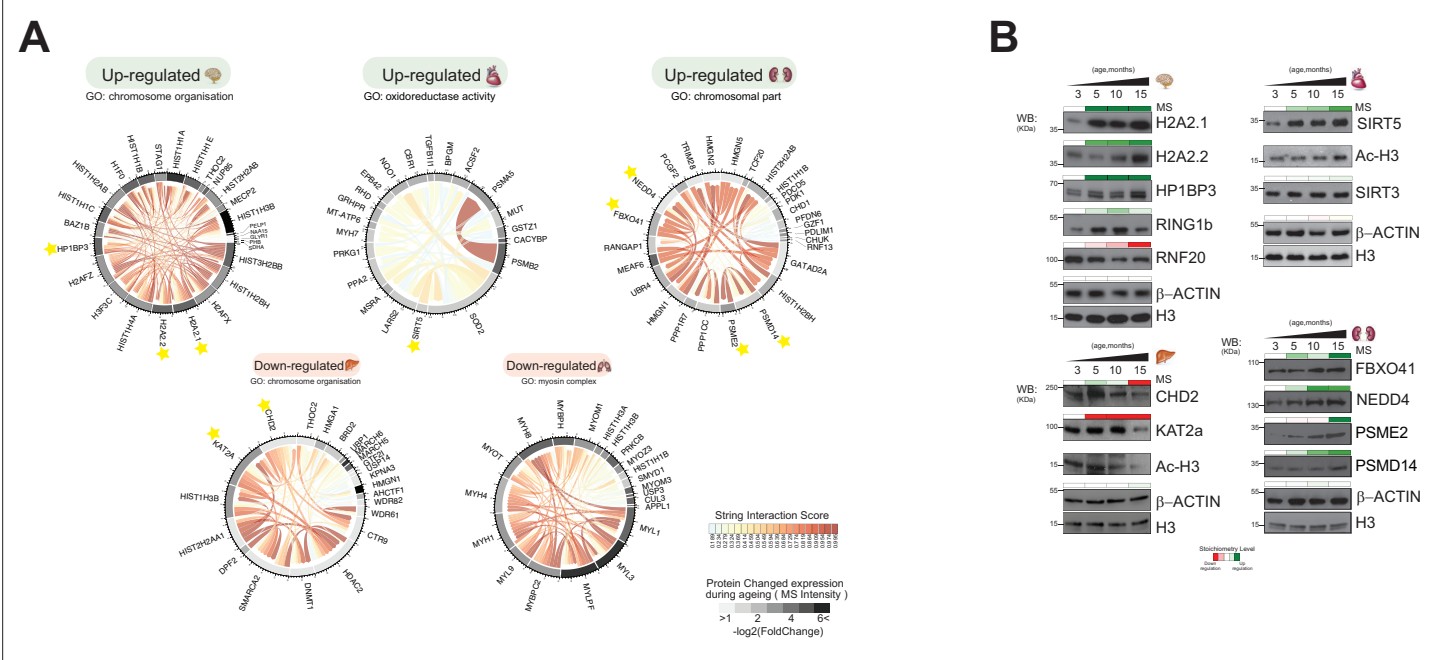

**Figure 7.** Distinct organ ageing profiles are defined by unique chromatin-associated proteins. (**A**) Protein interaction modules (obtained from the STRING database) are shown for Gene Ontology categories found to be significantly up- or downregulated in five organs. Each chord corresponds to a protein-protein interaction while the STRING interaction score is indicated by colour (red for high confidence). The quantitative differential protein expression during the mouse lifespan between 3 and 15 months is shown on the outer circle on a grey-black intensity scale. (**B**) Biochemical validation of four protein module responses to ageing identified using chromatin-associated proteomics. Organ lysates (from the brain, heart, liver, and kidney) were immunoblotted with the indicated antibodies. The bar above the blots corresponds to the quantitative protein expression levels determined in our proteomics experiments over time (green/red scale up- or downregulated, respectively).

The online version of this article includes the following source data for figure 7:

**Source data 1.** Original Western blotting data.

**Source data 2.** Original Western blotting data.

**Source data 3.** Original Western blotting data.

**Source data 4.** Original Western blotting data.

## Characterisation of regulated molecular networks in ageing mouse organs

We looked in more detail at the most significantly regulated organ-specific upregulated and down-regulated protein categories during mouse ageing. To validate the most significant organ-specific biological processes changing during mouse ageing, the selected GO terms related to chromatin environment were further investigated (*Figure 6*, right panel, dashed box). Additionally, we selected GO terms known to be associated with ageing effects to see whether chromatin-associated proteins fell into these categories.

We used protein-protein interaction (PPI) data from the STRING database to map the network of chromatin-associated protein belonging to the most significant GO categories of each organ (*Szklarczyk et al., 2015*; *Szklarczyk et al., 2017*; *Figure 7A*).

Subsequently, we combined the protein interaction networks with the quantitative dataset (*Figure 7A*). By integrating PPIs and protein expression, we derived co-interaction and co-expression networks to improve our understanding of biological mechanisms involved in ageing.

Finally, we attempted to confirm independently, by Western blot, the observations noted in our wider dataset, specifically to the co-expression network generated (*Figure 7B*).

### Brain
The majority of upregulated proteins of the ageing brain belonged to the category 'chromosome organisation', including histones and histone-binding enzymes (*Figure 7A*). We noticed a strong

subnetwork of PPIs between two histone variants and histone-binding enzymes: macro-H2A2.1, macro-H2A2.2, and HP1BP3.

Macro-H2A2.1, macro-H2A2.2, and HP1BP3 accumulated during mouse brain ageing (*Figure 7B*). Macro-H2A2.1, macro-H2A2.2, and HP1BP3 are mammalian heterochromatin components that are highly expressed in adult mouse brains (*Douet et al., 2017*; *Barrero et al., 2013*; *Garfinkel et al., 2015*; *Garfinkel et al., 2016*). All three proteins increased in expression during organ development and ageing. These proteins were not detected in any other mouse organ in this proteomics study (data not shown). Macro-H2A2.1 is an epigenetic marker whose major function is to maintain nuclear organisation and heterochromatin architecture (*Douet et al., 2017*). HP1BP3 is a heterochromatin marker protein that recognises the histone mark H3K9me3 and promotes transcriptional repression (*Garfinkel et al., 2015*). HP1BP3 loss of function is associated with cognitive impairment, suggesting a role for this protein in establishing or maintaining cognitive functions (*Garfinkel et al., 2016*). We also confirmed the expression of other heterochromatin markers, including RING1b (*Saksouk et al., 2015*) and RNF20 (*Kim et al., 2005*), used as protein controls to monitor our strategy (*Figure 7B*). These results suggest a link between regulation of heterochromatin components and accumulation of histone variants during the ageing of brain tissue in adult mammals.

## Heart

GO analysis of the regulated proteins of ageing heart tissue indicated upregulation of 'oxidoreductase activity' (*Figure 7A*). A chromatin-associated protein of the sirtuin family (SIRT5) is a member of this category. Sirtuins are histone deacylases that play an important role in age-related pathological conditions such as cancer and the deregulation of metabolism (*Gillette and Hill, 2015*; *Mei et al., 2016*; *Masse et al., 1991*). Sirtuin proteins are mostly annotated as mitochondrial proteins, but they can translocate further into the nucleus or other cell compartment (*Kupis et al., 2016*; *Vaquero, 2009*; *Li et al., 2013*). Our results suggest that the activation of specific members of the sirtuin family and their translocation to the nucleus is involved in the ageing process (*Kupis et al., 2016*; *Rajendran et al., 2011*).

SIRT5 and SIRT3 were detected in our chromatin-enriched proteome dataset of ageing mouse heart. Both SIRT3 and SIRT5 expression levels increased during the mouse lifespan, with SIRT5 being upregulated from 3 to 15 months (*Figure 7B*).

The abundance change of SIRT3 levels was less pronounced. This data was confirmed by Western blotting (*Figure 7B*). SIRT5 is a histone deacylase that removes malonyl, succinyl, and glutaryl groups from histones. The ageing-dependent increase in histone H3 acetylation observed in our proteomics study and by Western blotting (*Figure 7B*) is consistent with the fact that SIRT5 has no deacylase activity towards histone H3.

## Liver

We observed drastically downregulated 'chromosome organisation' in the liver during ageing (*Figure 7A*). This category contained histone acylation/acetylation-related chromatin remodelling enzymes associated with different chromatin-modifying complexes. Examples include KAT2A (either ATAC or SAGA complex), CHD2 (NuRD), HDAC2 (CoREST, NuRD, SWI/SNF, Sin3A-like), WDR82 (COMPASS), and DNMT1 (ACF) (*Medvedeva et al., 2015*; *Xu et al., 2016*; *Figure 7*).

Western blot analysis confirmed that the protein expression levels of CHD2 and KAT2A are significantly reduced at 15 months (*Figure 7B*), which leads to lower histone acylation levels. We indeed observed decreased levels of global histone H3 acetylation at this time point (*Figure 7B*), which contrasts with what we observe in the heart tissue. These results confirm our previously published data on decreased H3 acetylation (H3K14, K23, K27) in the liver tissue during ageing (*Tvardovskiy et al., 2017*), and that there is a decreased activity of histone acylation in the liver at late stages of ageing.

## Kidney

The upregulated 'chromosomal part' category in ageing kidney tissue included components of the SCF-type E3 ubiquitin-protein ligase family (FBXO41, RNF13, NEDD4, TRIM28) and the proteasome subunits PSMD14 and PSME2. (*Figure 7A*). The mechanistic links between proteasome activity and ageing are well established (*Saez and Vilchez, 2014*; *McCann and Tansey, 2014*).

The proteasome is a large self-compartmentalised protease complex that recognises, unfolds, and destroys ubiquitylated substrates (*McCann and Tansey, 2014*).

The protein expression levels of FBX041 and NEDD4 increased gradually during kidney ageing, while the signal intensity for PSMD14 and PSME2 was more pronounced at the latest time point (15 months) (*Figure 7B*). Thus, ageing kidney increases E3 ubiquitin-protein ligases, enhances the ubiquitylated substrates, and stimulates proteasome abundance and activity.

### Lungs

Many downregulated proteins of ageing lungs were involved in processes such as 'muscle organisation and reassembly', and the top GO term encompasses an array of myosin motor proteins (*Figure 7A*). The downregulated 'myosin complex' GO category included several proteins belonging to the myosin family, following the observations that human muscle ageing is accompanied by changes in the expression of myosins (*Murgia et al., 2017*). We did not perform any immunoblotting validation of these myosin proteins due to the high sequence similarity among myosin subunits and the lack of highly specific antibodies against them.

These observations suggested that ageing might influence the lung cell structure through alteration of the higher-order chromatin architecture.

### Spleen

Only a few differentially expressed proteins were observed in the spleen, and they did not allow for useful GO analysis and protein network analysis.

## Discussion

We implemented a comprehensive high-mass-accuracy mass spectrometry-based proteomics strategy to monitor changes in the chromatin-enriched proteomes of six mouse organs over a time course that mimics adult development and ageing, from the early adult to the mature adult stage.

Many age-related diseases are linked to changes in the transcriptional and epigenetic machinery that regulate gene expression.

We focused on the changes in the expression of proteins that mediate transcription, including DNA-binding proteins and chromatin modifiers such as 'writers', 'erasers', and 'readers' (*Signolet and Hendrich, 2015*; *Hyun et al., 2017*). Chromatin modifiers add, remove, or recognise particular PTMs of proteins associated with the alteration of chromatin architecture, and ultimately involved in the regulation of gene expression (*Santos and Lindner, 2017*).

Over 2000 proteins were quantified in each organ, generating a useful resource for researchers investigating mammalian development and ageing.

We identified distinct and organ-specific unique ageing features associated with each organ. We observed unique chromatin modifiers that were expressed and accumulated differently during ageing, leading to changes in the chromatin architecture, including changes in the expression of heterochromatin markers, histone deacetylases, ubiquitin-protein ligase, histone acetylation enzymes, and myosin complex in the brain, heart, kidney, liver, and lung, respectively.

Brain and spleen displayed the largest and most diverse and heterogenous chromatin proteomes. The brain is arguably the most complex organ of a mammal. Brain chromatin structure and function is sustained by a large set of chromatin-associated and nuclear proteins, which also exhibit temporal dynamics of expression during the mouse lifespan as demonstrated here. The spleen chromatin proteome was rather constant during the lifespan whilst large and diverse, which may reflect the physiological role of the spleen for continuous maintenance of important immune and defence activities of the organism.

We demonstrated progressive changes of chromatin-associated protein expression in response to ageing. Also, the specific nature of the organ was more of a significant discrimination factor, and subsequently distinct proteome profiles in response to ageing were observed. For instance, we noticed over the mouse lifespan strongly upregulated chromatin-associated proteins relate to distinctive pathways involved in 'oxidation-reduction response', 'response to oxidation stress' and 'nucleosome assembly', as well as signals that promote apoptosis processes. Conversely, chromatin-associated proteins

strongly downregulated are related to "muscle organisation and reassembly" and "histone-modifying enzymes" associated with chromatin assembly and organisation.

Our study of chromatin-enriched proteomes demonstrated that macro-H2A2 accumulates in the mouse brain during ageing. The epigenetic regulator HP1BP3 accumulates at a similar rate and very likely interacts with macro-H2A2. Both macro-H2A2 and HP1BP3 are highly expressed in the adult mouse brain, and we suggest that a complex involving these two proteins is implicated in maintaining heterochromatin integrity and promote gene silencing during the mouse lifespan.

Reversible acetylation of histones plays a critical role in transcriptional regulation in eukaryotic cells. We detected reduced levels of histone H3 acetylation during ageing in mouse liver. The opposite trend was observed in ageing heart, that is, an increase in histone H3 acetylation. Two families of deacylase enzymes were identified: the histone deacetylases, HDACs, and the Sir family protein (Silent Information Regulator)-like family of NAD-dependent deacylases, or sirtuins (*Grozinger et al., 2002*). Both enzyme families play a major role in gene regulation by modifying the histone acetylation/ acylation landscape in response to external stimuli and specific environmental stress conditions, such as oxidative stress (*Drazic et al., 2016*). In mammals, the brain and heart have the greatest oxygen demand for their ATP-dependent processes. Nearly all cellular processes of cardiomyocytes are driven by ATP-dependent pathways (*Hu et al., 2016*). SIRT3 plays an important role in maintaining basal ATP levels and regulated energy production in mouse embryonic fibroblasts (*Hu et al., 2016*; *Ahn et al., 2008*).

Sirtuin proteins are mostly annotated as mitochondrial proteins, but they can translocate into the nucleus or other cell compartments (*Kupis et al., 2016*; *Vaquero, 2009*; *Li et al., 2013*). The translocation of sirtuins through different cellular compartments is poorly described. Here, we speculate that the roles of SIRT3 and SIRT5 in the heart are essential as they compensate for age-related cellular dysfunction by controlling the levels of histone acylation marks. They may thereby promote the expression of proteins required for DNA repair to prevent cardiac hypertrophy in response to oxidative stress (*Hu et al., 2016*).

The SAGA complex is a multi-subunit histone-modifying complex. KAT2A is a SAGA component in mammals, containing both a HAT domain and a bromodomain and extended N-terminal domain which confers the ability to acetylate mononucleosomal H3 (*Gamper et al., 2009*). KAT2A is required for normal development in mice (*Xu et al., 2000*; *Lin et al., 2007*). Furthermore, KAT2A-level expression decreases during cell differentiation (*Xu et al., 2000*), suggesting that the downregulation of histone acetylation is connected with reducing activation of gene expression of target genes which promote self-renewal and pluripotency state during ageing. Overall, our experiments suggest a connection between the roles of two epigenetic enzymes CHD2 and KAT2A, whereby their mutual protein expression is associated with liver differentiation during the mouse lifespan.

The proteasome is a complex proteolytic machine formed by the assembly of several subunits (*Finley, 2009*). The ubiquitin-proteasome system (UPS) is the primary selective degradation system in eukaryotic cells, localised both in the nuclei and cytoplasm compartment, which is required for the turnover of soluble proteins (*Schmidt and Finley, 2014*). The UPS is mainly implicated in protein degradation in response to the regulation of several processes, including the maintenance of cellular quality control, transcription, cell cycle progression, DNA repair, receptor-mediated endocytosis, cell stress response, and apoptosis (*Lecker et al., 2006*). Before a protein is degraded, it is first flagged for destruction by the ubiquitin conjugation system, which ultimately results in the attachment of a polyubiquitin chain on the target protein (*Tanaka, 2009*; *Adams, 2003*).

Ubiquitin and the proteasome have been implicated in processes as diverse as the control of transcription, the response to DNA damage, the regulation of chromatin structure and function, and the export of RNAs from the nucleus (*McCann and Tansey, 2014*).

We found an increase in the levels of two proteasome subunits in ageing kidney organ consistent with increased E3-ubiquitin ligase activity. Increased expression of FBXO41, a subunit of the SCF E3 ubiquitin ligase complex, correlates with NEDD4 expression. Recent reports suggested a decline of proteasome function related to senescence observed in several mammalian tissues and human cells (*Bulteau et al., 2000*; *Carrard et al., 2003*; *Petropoulos et al., 2000*; *Bardag-Gorce et al., 1999*). During the ageing process, dysfunction of the ubiquitination machinery or the proteolytic activity may occur, leading to proteasome failure, which is linked to several age-related human diseases (*Schmidt and Finley, 2014*; *Chondrogianni and Gonos, 2010*). We, therefore, speculate that an increased

expression of E3-ubiquitination ligase activity may compensate for the proteasome activity during ageing in the kidneys.

Not all functions of actin and actin-related proteins in complexes are yet clear: it is known that they play important roles in maintaining the stability of the proteins, possibly by bridging subunits and recruiting the complexes to chromatin (*Farrants, 2008*). In line with previous analysis, the majority of downregulated 'chromatin-associated proteins' in lung tissue were assigned to the categories 'myosin complex', 'chromatin organisation' and 'histone modification'. The most significant subnetwork corresponds to the 'muscle organisation and reassembly' category and is related to the myosin family. This is in accordance with recent reports where the change of protein expression of particular myosin subsets implied a human ageing response (*Murgia et al., 2017*; *Lang et al., 2018*) The presence of actomyosin-like protein in the chromatin environment raises questions about the role of actin-like protein such as myosin in nuclear and chromatin processes (*Walther and Mann, 2011*).

In the spleen, we detected a large number of chromatin-associated proteins but only a fraction of these proteins were differentially regulated during ageing. Thus, the spleen chromatin proteome exhibits a 'young' or 'age-less' phenotype. Consequently, only a few pathway-level changes were observed, possibly indicating the important role of this organ in maintaining immune functions, removing old red blood cells and microbes which seem, from our data, to be not affected by ageing and consistent with our hypothesis that the time-course changes in protein expression distinctly affect each organ. We observed the largest numbers of chromatin remodelling proteins in the spleen, which suggests that this organ is may provide a good model for epigenetic studies.

Overall, our findings using high mass resolution LC-MS/MS suggest a new approach to investigate the dynamic chromatin protein environment during the lifespan of an organism. We provide a high-quality and robust dataset of protein expression changes in mouse organs during the ageing process. The dataset shows that in vivo models can describe how the dynamic changes of chromatin-associated protein may alternatively promote or repress gene expression during ageing, also reflecting some physiological features of the organ.

Our study adds novel details to mouse biology and chromatin dynamics of organs, and it complements previous attempts to identify biomarkers for mouse lifespan (*Shavlakadze et al., 2019*; *Rappsilber et al., 2003*). Walther et al. reported that the bulk proteins' abundance is less prone to changes in organs such as the brain, heart, and kidney obtained from mice aged 5 or 26 months. They reported only a few proteins that exhibited statistically significant expression changes during ageing (*Shavlakadze et al., 2019*). Thus, bulk proteome analysis of mammalian organs has limitations, whereas organelle-specific proteomics, as presented here for chromatin, is a more viable strategy to reveal the molecular details of important biological processes, such as ageing and chromatin regulation.

Using a rat model (*Rappsilber et al., 2003*), Glass et al. studied the alterations of gene expression to identify a putative mammalian ageing signature. Unfortunately, our chromatin proteomics strategy does not readily compare to the study by Glass et al. as only three out of the seven time points are in common between the studies. We observed common ageing signatures (trends) such as cell stress response and transcriptional alterations changing at the late stage of adult rodent lifespan.

We see a consistent overlap between our results and a list of human ageing-related biomarker candidates of the 'Human Ageing Genomic Resources' and 'GenAge machine learning databank' (*Figure 5*; *Tacutu et al., 2013*; *Kerepesi et al., 2018*). Our differentially expressed candidate mouse proteins behaved just as human ageing biomarkers or ageing-related human proteins that promote disease.

Whereas our study provides novel features and details of molecular ageing processes in mammals, it does not provide the mechanistic details of the protein-mediated ageing process in chromatin. Quantitative proteomics is an important tool for further studies of chromatin dynamics, and the emerging field of high-sensitivity single-cell proteomics will assist in revealing the features of organ function in health, ageing, and disease using limited sample amounts. The experimental protocols used in this study provide a foundation for a more detailed interrogation of chromatin biology by functional proteomics. The data resource associated with this study provides a framework for generating novel hypotheses aimed at revealing the molecular features of ageing and developing novel approaches to mitigate age-related ailments.

# Materials and methods

**Key resources table**

| Reagent type (species) or resource | Designation | Source or reference | Identifiers | Additional information |
|---|---|---|---|---|
| Antibody | HRP goat anti- mouse (mouse polyclonal) | Merck | Cat# 401253; RRID:AB_437779 | WB (1:10,000) |
| Antibody | HRP goat anti-rabbit (rabbit polyclonal) | Sigma | Cat# A0545; RRID:AB_257896 | WB (1:10,000) |
| Antibody | HRP anti-guinea pig (guinea pig polyclonal) | Gift from Benny Garfinkel | | WB (1:10,000) |
| Antibody | Anti-SIRT3 (rabbit monoclonal) | Cell Signaling Technology | Cat# 2627S; RRID:AB_2188622 | WB (1:1000) |
| Antibody | Anti-NEDD4 (rabbit polyclonal) | Cell Signaling Technology | Cat# 2740S; RRID:AB_2149312 | WB (1:500) |
| Antibody | Anti-PSME2 (rabbit polyclonal) | Cell Signaling Technology | Cat# 2409S; RRID:AB_2171085 | WB (1:500) |
| Antibody | Anti-SIRT5 (rabbit monoclonal) | Cell Signaling Technology | Cat# 8782S; RRID:AB_2716763 | WB (1:1000) |
| Antibody | Anti-CHD2 (rabbit polyclonal) | Cell Signaling Technology | Cat# 4170S; RRID:AB_10621947 | WB (1:500) |
| Antibody | Anti-KAT2A/GCN5 (rabbit monoclonal) | Cell Signaling Technology | Cat# 3305; RRID:AB_2128281 | WB (1:500) |
| Antibody | Anti-PSMD14 (rabbit monoclonal) | Cell Signaling Technology | Cat# 4197S; RRID:AB_11178935 | WB (1:1000) |
| Antibody | Anti-RING1B/RNF2 (rabbit monoclonal) | Cell Signaling Technology | Cat# 5694P; RRID:AB_10705604 | WB (1:500) |
| Antibody | Anti-H3 (rabbit polyclonal) | Abcam | Cat# ab1791; RRID:AB_302613 | WB (1:20,000) |
| Antibody | Anti-macroH2A.1 (rabbit polyclonal) | Abcam | Cat# ab37264; RRID:AB_883064 | WB (1:1000) |
| Antibody | Anti-macroH2A2.2 (rabbit polyclonal) | Abcam | Cat# ab4173; RRID:AB_2115110 | WB (1:1000) |
| Antibody | Anti-RNF20 (rabbit polyclonal) | Cell Signaling Technology | Cat# 9425S; RRID:AB_2797700 | WB (1:1000) |
| Antibody | Anti-H3acetyl (rabbit polyclonal antibody) | Active motif | Cat# 39139; RRID:AB_2687871 | WB (1:500) |
| Antibody | Anti-beta actin (mouse monoclonal) | Sigma | Cat# A5441; RRID:AB_476744 | WB (1:5000) |
| Antibody | Anti-FBX041 (rabbit polyclonal) | Protein Tech | Cat# 24519-1-AP; RRID:AB_2879586 | WB (1:500) |
| Antibody | Anti-HP1BP3 (guinea pig polyclonal) | Gift from Benny Garfinkel | | WB (1:2000) |
| Software, algorithm | Xcalibur software | Thermo Scientific | RRID:SCR_014593 | |
| Software, algorithm | Progenesis QI v2.2 | Waters | RRID:SCR_018923 | |
| Software, algorithm | Proteome Discoverer v2.1.0.81 | Thermo Scientific | RRID:SCR_014477 | |
| Software, algorithm | PolySTest | | RRID:SCR_021942 | http://computproteomics.bmb.sdu.dk:8192/app_direct/PolySTest/ |

## Animals and organ collection

Male C57BL/6J mice were obtained from a study approved by the Danish Animal Ethics Inspectorate (J.nr. 2011/561-1950). Wild-type mice were bred in the Biomedical Laboratory, University of Southern Denmark, under a 12 hr/12 hr light/dark cycle (lights on at 6:30 am). Food and water were available

ad libitum. Mice were sacrificed by cervical dislocation at the ages of 3, 5, 10, and 15 months. Liver, kidneys, brain, heart, spleen, and lungs were excised, rinsed in ice-cold phosphate-buffered saline (PBS), and immediately snap frozen. Organs were stored at –80°C until further processing.

## Isolation of chromatin lysate in mouse embryonic cells (ESCs)

Mouse embryonic cell pellets were washed in PBS and resuspended in lysis buffer (25 mM Tris·HCl pH 7.6, 150 mM NaCl, 1% NP40, 1% sodium deoxycholate, 0.1% SDS, cOmplete Protease Inhibitor w/o EDTA [Roche], 0.5 mM DTT). The lysates were incubated for 15 min on ice and cell membranes disrupted mechanically by syringing five times with 21G narrow gauge needle and sonicating 3 Å approximately for 2 s at high power. Lysates were incubated on ice for another 15 min and cleared by centrifugation at 14,000 rpm 4°C 30 min. To harvest the nuclear fraction, lysates were resuspended in an equal volume of Nuclear Buffer (120 mM NaCl, 20 mM HEPES pH 7.9, 0.2 mM EDTA, 1.5 mM MgCl$_2$, 20% glycerol, cOmplete Protease Inhibitor w/o EDTA [Roche], 0.5 mM DTT) and dounced 20 times with tight pestle type B. Lysates were incubated for 45 min rotating to dissociated chromatin-bound proteins and pre-cleared by centrifugation at 14,000 rpm 4°C for 30 min. Subsequently, nuclear pellets were lysed in buffer C containing protease inhibitors (420 mM NaCl, 20 mM HEPES pH 7.9, 20% [v/v] glycerol, 0.42 M NaCl, 1.5 mM MgCl$_2$, 0.2 mM EDTA, cOmplete Protease Inhibitor w/o EDTA [Roche], 0.5 mM DTT). Lysates were incubated for 1 hr rotating at 4°C, in the presence of 250 U/mL benzonase nuclease, to form dissociated chromatin-bound proteins and pre-cleared by centrifugation (20,000 × $g$, 1 hr at 4°C). After centrifugation, the supernatant was snap frozen.

## Isolation of chromatin lysate in mouse organ

Organ samples were homogenised on ice in a homogenisation buffer (2.2 M sucrose, 10 mM HEPES/ KOH pH 7.6, 15 mM KCl, 2 mM EDTA, 0.15 mM spermine, 0.5 mM spermidine, 1 mM DTT, and cOmplete Protease Inhibitor w/o EDTA [Roche]), 0.5 mM PMSF, and phosphatase inhibitor (PhosSTOP, Roche) using a 'loose'-type pestle tissue. The solution was stacked over with a cushion buffer (2.05 M sucrose, 10 mM HEPES/KOH pH 7.6, 15 mM KCl, 2 mM EDTA, 0.15 mM spermine, 0.5 mM spermidine, 1 mM DTT, cOmplete Protease Inhibitor w/o EDTA [Roche], 0.5 mM PMSF) in an ultracentrifuge tube and cleared by centrifugation (20,000 × $g$, 1 hr at 4°C). Pellet containing nuclei was washed twice with 1 mL Dulbecco PBS (3000 × $g$, 5 min at 4°C). To harvest the nuclear fraction, lysates were subsequently resuspended in buffer C (420 mM NaCl, 20 mM HEPES pH 7.9, 20% v/v glycerol, 2 mM MgCl$_2$, 0.2 mM EDTA, 0.1% NP40, cOmplete Protease Inhibitor w/o EDTA [Roche], 0.5 mM DTT) and dounced 20 times with a 'tight' pestle. Lysates were incubated for 1 hr rotating at 4°C, in the presence of 250 U/mL benzonase nuclease, to form dissociated chromatin-bound proteins and pre-cleared by centrifugation (20,000 × $g$, 1 hr at 4°C). After centrifugation, the supernatant was snap frozen.

## Immunoblotting

Protein lysate was quantified by using the Bradford assay. Approximately 30 µg protein lysates were separated on SDS–PAGE gels and transferred to nitrocellulose membranes. Membranes were blocked with 5% non-fat milk or 5% BSA at room temperature for 1 hr and incubated overnight with diluted primary antibody at 4°C. Membranes were then washed and incubated with HRP-conjugated goat-anti-rabbit or mouse IgG secondary antibody for 1 hr at room temperature. Membrane was incubated with enhanced chemiluminescence reagents (Thermo Scientific) followed by exposure to X-ray films. Immunoblotting was performed using the antibodies and conditions listed in the Key resources table.

## Protein sample preparation for mass spectrometry

Protein concentration was measured by using the Bradford assay. 50 µg of chromatin lysate was precipitated with 4 vol of acetone and resuspended in 20 µL 6 M urea/2 M thiourea in 50 mM ABC. Cysteines were reduced with 1 mM DTT for 30 min and alkylated with 5 mM iodoacetamide for 30 min. 500 ng LysC (Wako) was added, and proteins were digested at room temperature for 3 hr. Samples were diluted 1:4 with 50 mM ABC and 1 µg trypsin (Sigma) and incubating overnight at 37°C. Digestion was quenched by addition of formic acid (FA) to a final concentration of ~1%, and the resulting peptide mixture was centrifuged at 5000 × $g$ for 15 min. Peptides in the supernatant were desalted by stage tipping with C18 material (*Silva et al., 2006*) and eluted by 50% ACN, 0.1% FA in water, and dried in a SpeedVac. Samples were stored at –80°C until further use.

## Mass spectrometry analysis

Samples were redissolved in 50 µl of trifluoroacetic acid 0.1% (vol/vol) in water, as buffer A, and sonicated for 1 min and centrifuged for 15 min at 15,000 × *g*. Analysis was carried out on an Ultimate 3000 RSLCnano HPLC system connected to a mass accuracy high-resolution mass spectrometry, Q Exactive HF (Thermo Fisher). The MS instrument was controlled by Xcalibur software (Thermo Fisher). The nano-electrospray ion source (Thermo Fisher) was used with a spray voltage of 2.2 kV. The ion transfer tube temperature was 275°C. Samples were loaded on a cartridge pre-column PepMap 100 5 * 0.3 mm (Thermo Fisher) in 2% ACN, 98% $H_2O$, 0.1% TFA at 10 µL/min, and then separated either with an easy Spray C18 or a 75 µm * 50 cm 2 µm PepMap 100 column (Thermo Fisher). Separation was done in a linear gradient of ACN, 0.1% FA (buffer B) in $H_2O$, 0.1% FA (buffer A, from 4% to 32% B in 2 hr) at 0.25 µL/min at 45°C. To avoid sample carryover between different samples, both pre-column and column were washed with 3 * 10 min gradients from 2% to 95% B (3 min at 2% B – 3 min from 2% to 95% B – 3 min at 95% B – 1 min from 95% to 2% B). MS analysis was done in DDA mode with 1 MS1 scan, followed by 20 dependent MS2 scans. MS1 parameters were 120,000 resolution, 3e6 AGC target, maximum IT 100 ms, with a scan range of 300–2000 *m/z*. MS2 parameters were 15,000 resolution; 2e5 AGC target; maximum IT 15 ms; isolation window 1.2 *m/z*; isolation offset 0 *m/z*; fixed first mass 110 *m/z*; (N) CE 30; minimum AGC 8e3; exclude unassigned and 1; 6–8 charges; preferred peptide match; exclude isotopes, and dynamic exclusion was set to 40 s. All mass spectrometry raw data were deposited in MassIVE (https://massive.ucsd.edu/ProteoSAFe/static/massive.jsp) with accession numbers MSV000084270, MSV000084279, and MSV000084375.

## Data processing

A combination of Progenesis QI v2.2 (Waters) and Proteome Discoverer v2.1.0.81 (Thermo Scientific) was used to estimate the relative protein abundance of protein and peptide using a label-free approach. Thermo Raw MS files were imported into Progenesis QI v2.2 (Waters) and the match-between-runs feature was enabled. Subsequently, the matching and alignment time window were performed according to *m/z* and retention time enable using default settings. Filtering only ions with a charge state of up to +4 was considered. The aligned ion intensity map was carried out using default Peak Peaking settings, and no further imputation analysis was performed. The aligned ion intensity map was exported in .pepXml files and imported into Proteome Discoverer v2.1.0.81 (Thermo Scientific) for further protein and peptide identification and searched against the SwissProt mouse reference database by using an in-house MASCOT server (v2.5.1, Matrix Science Ltd, London, UK). Database searches were performed with the following parameters: Fixed Mod: cysteine carbamidomethylation; Variable Mods: methionine oxidation; Trypsin/P digest enzyme (maximum two missed cleavages); Precursor and fragment mass tolerance was set to 10 ppm and 0.8 Da, respectively. Identified peptides and proteins were filtered using a false discovery rate (FDR) set at 1% and a peptide spectrum match (PSM) set at 1%. Subsequently, the MS/MS and ion abundance search was exported in .mgf files and imported into Progenesis QI v2.2 to perform peptide and protein normalisation and relative quantitation using respectively a label-free analysis Hi-N/3 summarisation method and ion abundance normalisation default method (*Ferguson et al., 2005*; *Korthauer et al., 2019*; *Supplementary file 3*, Table S3, *Figure 5—figure supplement 1B*). In terms of proteomic depth and data management, for each biological sample two technical replicates were combined by summing both protein ion abundance values. Due to the large proteome scale, each organ were analysed separately to maintain the reliability of the statistical analysis and prevent potential homogenisation of the protein expression differences that occurred during the mouse's lifespan. Protein group databases are listed in *Supplementary file 3*, Table S3.

## Quantitative analysis and interpretation

To identify the proteins that exhibited significant abundance changes during ageing, each organ dataset was further normalised against the 'core proteome', that is, the proteins identified in all six organs over all four time points. Subsequently, statistical tests based on the rank product test were carried out to quantify the dynamic protein expressions changing during the mouse lifespan (*Koziol, 2010*; *Breitling et al., 2004*; *Figure 5—figure supplement 2A*). FDRs were calculated to correct for multiple testing (*Tyanova et al., 2016*).

Hierarchical heatmap clustering based on Euclidean distance was performed using Perseus software (1.6.0) (*Tyanova et al., 2016*). Z-scores were calculated by subtracting the mean of protein abundance values in all samples and dividing by the standard deviation. Protein group tables are listed in *Supplementary file 4*, Table S4.

GO analysis of differentially expressed proteins was performed using the DAVID online tool (v6.8) to obtain the biological processes (BPs) and pathways from the enriched chromatin proteome organs during ageing. FDR 1% was set as the minimum threshold value. Each dataset was independently analysed, and the *M. musculus'* (mouse) genome/proteome was used as background for each organ source. To generate a detailed dataset, which provides information on the location and topology of the protein in the cell, the data was sorted according to the Uniprot (SwissProt) subcellular location library. In particular, the subcellular location section was upgraded with the most recent depository protein libraries relative to chromatin studies (*Medvedeva et al., 2015*; *van Mierlo et al., 2019*; *Christoforou et al., 2016*; *Xu et al., 2016*).

## Acknowledgements

We thank professor Bente Finsen, Institute of Molecular Medicine, University of Southern Denmark, Odense, Denmark, for providing C57Bl6 mice. We also acknowledge Dr. Benjamin P Garfinkel (Harvard TH Chan School of Public Health), for the kind gift of HP1BP3 antibody for the immunoblotting experiment. We also thank Aliaksandra Radsizheuskaya (The Memorial Sloan Kettering Institute, New York) for providing mouse embryonic cell pellets for the initial chromatin-proteome experiment. SK and AR-W were supported by a grant from the Independent Research Fund Denmark—Natural Sciences (to ONJ). GO was supported by a grant to the Center for Epigenetics from the Danish National Research Foundation (DNRF #82). Proteomics and mass spectrometry research at SDU are supported by generous grants to the VILLUM Center for Bioanalytical Sciences (VILLUM Foundation grant no. 7292 to ONJ) and PRO-MS: Danish National Mass Spectrometry Platform for Functional Proteomics (grant no. 5072-00007B to ONJ).

## Additional information

### Funding

| Funder | Grant reference number | Author |
| --- | --- | --- |
| Villum Fonden | 7292 | Ole N Jensen |
| Danmarks Grundforskningsfond | DNRF #82 | Ole N Jensen |
| Danish Agency for Science and Higher Education | 5072-00007B | Giorgio Oliviero |

The funders had no role in study design, data collection and interpretation, or the decision to submit the work for publication.

### Author contributions

Giorgio Oliviero, Conceptualization, Investigation, Methodology, Visualization, Writing – original draft, Writing – review and editing; Sergey Kovalchuk, Methodology, Writing – review and editing; Adelina Rogowska-Wrzesinska, Conceptualization, Methodology, Writing – review and editing; Veit Schwämmle, Conceptualization, Formal analysis, Methodology, Writing – review and editing; Ole N Jensen, Conceptualization, Funding acquisition, Project administration, Resources, Supervision, Writing – review and editing

### Author ORCIDs

Adelina Rogowska-Wrzesinska http://orcid.org/0000-0002-9876-0061
Ole N Jensen http://orcid.org/0000-0003-1862-8528

## Ethics

Male C57BL/6J mice were obtained from a study approved by the Danish Animal Ethics Inspectorate (J.nr. 2011/561-1950).

## Decision letter and Author response

Decision letter https://doi.org/10.7554/eLife.73524.sa1
Author response https://doi.org/10.7554/eLife.73524.sa2

---

## Additional files

### Supplementary files

- Supplementary file 1. Supplemental Table S1.
- Supplementary file 2. Supplemental Table S2.
- Supplementary file 3. Supplemental Table S3.
- Supplementary file 4. Supplemental Table S4.
- Supplementary file 5. Supplemental Table S5.
- Transparent reporting form
- Source data 1. Raw and annotated Western blot images.

### Data availability

Proteomics data is deposited in public repository as specified in manuscript.

The following datasets were generated:

| Author(s) | Year | Dataset title | Dataset URL | Database and Identifier |
|---|---|---|---|---|
| Oliviero G, Kovalchuk S, Rogowska-Wrzesinska A, Schwämmle V, Jensen ON | 2021 | Enrichment of chromatin associated protein in mESC using LC-MS/MS | https://massive.ucsd.edu/ProteoSAFe/static/massive.jsp | MSV000084270, MSV000084270 |
| Oliviero G, Kovalchuk S, Rogowska-Wrzesinska A, Schwämmle V, Jensen ON | 2021 | Shotgun proteomics deciphered age pathways in mammalian organisms. BRAIN,TOTAL LYSATE, AGING | https://massive.ucsd.edu/ProteoSAFe/static/massive.jsp/MSV000084279 | MSV000084279, MSV000084279 |
| Oliviero G, Kovalchuk S, Rogowska-Wrzesinska A, Schwämmle V, Jensen ON | 2021 | Shotgun proteomics all mouse tissues | https://massive.ucsd.edu/ProteoSAFe/static/massive.jsp/MSV000084375 | MSV000084375, MSV000084375 |

---

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
