## [Editor Report]

The authors have performed an extensive analysis of chromatin enriched proteins as a function of age in mice. Time-course quantitative proteomics reveals the molecular complexity and diversity of mammalian organs and identified aging-related molecular features of chromatin.

---

## [Decision Letter]

[Editors' note: this paper was reviewed by Review Commons.]

---

## [Author Response]

Reviewer 11. The supplementary data was missing, so it is difficult to look at the quality of the data itself.

We apologize for this oversight. We shared with the reviewers the mass spectrometry RAW data deposited on MassIVE:

LINK-1- mESC: ftp://massive.ucsd.edu/MSV000084270/

LINK-2- Tissue Brain Total Lysate: ftp://massive.ucsd.edu/MSV000084279/

LINK-3- Tissue six organs Chromatin Lysate: ftp://massive.ucsd.edu/MSV000084375/

We also attach the large dataset from the ProteinGroup file of our mass spectrometry analysis to the manuscript.

In the supplementary material we updated all the ProteinGroup.file (Proteome Tables) in excel format. They correspond to:

1. Proteome of mouse embryonic stem cell obtained from whole cell lysate and chromatin enriched lysate.

File name: Supplementary material, Table S1

2. Brain proteome whole cell lysate during adult mouse lifespan

File name: Supplementary material, Table S2

3. proteomics of six mouse organs over time

File name: Supplementary material, Table S3

4. Uniquely regulated proteins organ-specific during ageing.

File name: Supplementary material, Table S4

2. Table 2: please remove the sum of all proteins which is not helpful. The combined number (currently in parentheses) is all we need. Please add the number of unique proteins for each organ.

We adjusted Table 2 as suggested.

3. It is interesting that the spleen has less changes over age than other organs. the authors may want to discuss this considering the differences in longevity of cell types.

We agree. We included a more detailed presentation of the spleen results in the manuscript and made this observation more prominent in the abstract, results and Discussion sections.

4. Generally, there is quite a bit of information missing in many figure legends or the figures themselves. Some examples are below. Please check carefully.

Thanks for this important comment. We have revised the legends and figures accordingly.

5. Please provide details about statistical analysis of proteomics data. Was ANOVA used? Was data FDR controlled?

We revised the description of data analysis (material and method section), line 843

The proteome dataset was obtained from organ samples harvested during adult mice lifespans. We focused on proteins that changed in one direction during the ageing of the organ, or linearly throughout the lifespan of the animal. A statistical strategy was applied to detect the most significant chromatin proteome changes associated with ageing. We used rank product tests to identify proteins that exhibited significant abundance changes during ageing (supplementary material, Figure 5, Panel D). We called these “differentially regulated proteins”. Depending on the experiment’s intrinsic data structures and variations, rank product tests was applied (DOI: https://doi.org/10.1074/mcp.RA119.001777). Significance thresholds were then calculated via a permutation-based false discovery rate using adjusted q-values calculated to correct for multiple testing (-log10 qValue < 0.1 cutoff)

6. I am not aware how Progenesis works. Is there some form of imputation when there are missing values?

We revised the description of data analysis (material and method section):

A combination of Progenesis QI v2.2 (Waters) and Proteome Discoverer v2.1.0.81 was used to estimate the relative protein abundance of protein and peptide using in label-free approach. For a better estimation and comparison of protein expression across different biological conditions, the Progenesis QI v2.2 integrated the MS files according to *m/z* and retention time features (Peak Peaking Strategy). The outcome increased the reliability and precision of peptide abundance measurements. After a Peak Peaking strategy was applied, no further imputation analysis was performed. The aligned ion intensity map was exported in.pepXml files and imported into Proteome Discoverer v2.1.0.81 (Thermo Scientific) for further protein and peptide identification.

By, Progenesis QI peptide ion abundance was normalised using the default method (Ferguson et al., 2005: "Housekeeping proteins: a preliminary study illustrating some limitations as useful references in protein expression studies", DOI: 10.1002/pmic.200400941). Progenesis QI for proteomics uses ratiometric data in log space, along with a median and mean absolute deviation outlier filtering approach, to calculate the scalar factor. This is a more robust approach, which is less influenced by noise in the data and any biases owing to abundant species, as the absolute values of abundance are disregarded.

7. Please explain better the heatmap/clustering. I assume the data represents z-scores? What are the colours to the right of the heatmaps?

We revised the heatmaps and clustering in Figure 5 and revised the description of data analysis line 848.

Hierarchical heatmap clustering based on Euclidean distance was performed using Perseus software (1.6.0) (100). Z-scores were calculated by subtracting the mean of protein abundance values in all samples and dividing by the deviation.

The colours on the right of side of the heatmaps indicate blue chromatin associated protein, green nuclear protein, and yellow protein associated with other cellular components. Description present in the figure legend.

8. Figure 5c, please provide the individual months in the bar graphs. To what has the data been normalised?

Figure 5c was adjusted following reviewer’s suggestions.

Reviewer 21. The link to the "mass spectrometry raw data" just has "XXX" as the location of the data. The processed data table is also not shared; what are the peptide and protein quantifications for every gene at every timepoint? The value of this study is largely because it will serve as a useful reference for future studies, but the data are obscured; figure 5C is the only place we actually see the fold changes across age for any named proteins. Or did I miss this somewhere? I looked through twice as I was surprised such a critical piece of information was missing.

We apologize for this omission.

We share the mass spectrometry RAW data deposited on MassIVE:

LINK-1- mESC: ftp://massive.ucsd.edu/MSV000084270/

LINK-2- Tissue Brain Total Lysate: ftp://massive.ucsd.edu/MSV000084279/

LINK-3- Tissue six organs Chromatin Lysate: ftp://massive.ucsd.edu/MSV000084375/

We also attach the large dataset from the ProteinGroup file of our mass spectrometry analysis to the manuscript.

In the supplementary material we updated all the ProteinGroup.file (Proteome Tables) in excel format. They correspond to:

1. Proteome of mouse embryonic stem cell obtained from whole cell lysate and chromatin enriched lysate.

File name: Supplementary material, Table S1

2. Brain proteome whole cell lysate during adult mouse lifespan

File name: Supplementary material, Table S2

3. Chromatin proteomics of six mouse organs over time

File name: Supplementary material, Table S3

4. Uniquely regulated proteins Organ-specific during ageing.

File name: Supplementary material, Table S4

2. In the introduction, the authors state that "one adult mouse month equivalent to approximately three human years".

At note (2) the reviewer points out the lack of reference in the following sentence from the Introduction paragraph: “Mice have a relatively short lifespan, with one adult mouse month equivalent to approximately three human years”. We agree with the reviewer and we have added the relevant reference.

The two articles from Pallav Sengupta and co-authors attempt to compare rodent and human age at different phases of their life. The precise correlation between the age of laboratory rodents and humans is still a subject of debate. Both articles try to fill this gap between approximation and accuracy, and they suggest a precise relation between murine and human age using multiple assessments central to a comparison of phenotypes and physiology features.

3. Figure 2C – can you also add a histogram for "intersect of all six tissues"? A Venn diagram would be overly complex here but you can use the "UpSetR" R package to make a comprehensible histogram showing overlap. Ok, I wrote this comment before reading the rest of the paper – I see you already did that. I think Figure 4A can already go directly here in Figure 2C? We don't really need to go through the analysis of Figure 3 to get here.(5) Table 2 and Figure 2C are redundant; just color Figure 2C the same way with the blue/green/yellow and then the ratios will be clear without the need for the table. Table 3 is similarly redundant with Figure 4A.

Reviewer #2 suggested some editing of Figure 2 and Table 2 to avoid redundancy between tables and figures in the manuscript. We thank Reviewer #2 for the advice, but we are also aware that those changes may affect the meaning of Figure 2. We therefore maintain Figure 2 and Table 2 in order to maintain consistency in the manuscript. We rephrased the manuscript text in order to avoid any redundancy or misleading information between Figure 2C and Table2. In the paragraph “Quantitative chromatin proteomics of ageing mouse organs”, Figure 2 showed the overall number of proteins identified across six organs over time, while Table 2 highlighted a more in-depth meaning to Figure 2. Table 2 showed the proteins detected in six mouse organs across time and displayed the number of annotated proteins associated with their subcellular location. Table 2 also showed the number of unique proteins in each organ for each subcellular location, suggesting a difference at the protein level appearing across all organs.

4. Why does Figure 3B only show one square per condition? Is this the average of the three replicates per time? Are these biological replicates in Figure 3A, or technical? (I assume biological, but I don't see it unambiguously stated.)

We adjusted Figure 3 Panel B by introducing the biological replicates per time point in the Pearson correlation coefficients cluster, line: 238. Previously the average of each biological replicate across each time point was included in the analysis. Reviewer #2 pointed out an interesting observation and we adjusted the figure by introducing all the biological replicates for all samples across all time points. We demonstrated reproducibility of each biological replicate and confirmed the robustness of our biochemical and proteomics methodology. We observed that this round of analysis did not differ from the previous analysis. The Pearson correlation coefficients Cluster (figure 3B) showed the consistency and reproducibility of analysis of three biological replicates at all time points and revealed distinct organ proteome profiles. We thank Reviewer #2 for the suggestions which improves the quality of Figure 3B.

Notes (6), (7), (8) and (13) from Reviewer #2 focused on the GO term analysis. Notes (6), (7) and (8) corresponded to Figure 4C and note (13) related to Figure 6. We thank Reviewer #2 for the constructive feedback providing useful insights that contributes to the GO term analysis in the manuscript.

6. The authors mention that "the core chromatin proteome contained proteins associated with the major transcriptional/epigenetic chromatin complexes such as…" which I don't doubt, but the way it is presented it could also just be cherry-picking. Please run the protein composition through DAVID to show whether those categories of proteins are enriched. Make sure to also upload a specific "Background" list to DAVID that includes only the proteins measured, as by default it will compare your uploaded list against all genes, which is not accurate for proteomics since the 863 measured proteins are going to be a biased sample. Please do the same thing for every subsequent list of genes that is listed, e.g. the "they included the histone binding protein of the methyltransferase category…" for brain in the following paragraph, spleen in the following paragraph, etc. For each, please include a supplemental table with the genes uploaded for each DAVID analysis as well as the different background list for each analysis. OK, I'm reading sequentially while I write this review, and I see such a GO analysis was done later on the page, but the analysis should be included directly with the first time the list of proteins are mentioned (e.g. lines 280-288 should go directly after line 252).

We re-edited the manuscript in order to avoid any confusion about the GO term analysis performance, line: 835. In the “Material and Method” section, <<the Gene ontology (GO) analysis of differentially expressed proteins was performed using the DAVID online tool (v6.8) to obtain the biological processes (BPs) and pathways from the enriched chromatin proteome organs during ageing. FDR 1% was set as the minimum threshold value>>.

7. The GO analysis results don't always agree with the earlier text. For instance Figure 4B agrees with the text (i.e. that translation genes are enriched in the common 863), but figure 4C with the brain proteins does not list any methyltransferase or histone deacetylase category-or indeed anything related to translation. Thus it seems from the GO analysis that the genes selected in lines 259-260 were cherry picked rather than representatives of a statistically significant cluster.

In order to investigate a comprehensive functional analysis of the proteomics datasets across all the organs during ageing we applied an UpSetR plot cluster analysis. We first observed a shared proteome, with 863 proteins present across all organs over time (Figure 4, Panel A, Table 3). We labeled the 863 shared proteins as the “Core” proteome. As expected from the overlapping of six proteome organs across time, functional analysis of the “Core” proteome anticipated the presence of categories associated with aging response, such as “response to stress”, “system development”, “cell cycle”, and “regulation of cell death”. Since our strategy was performed in the chromatin and nuclear environment the resulting GO term categories matched with chromatin cellular environments, such as “nucleosome assembly”, “chromatin remodeling”, and “covalent chromatin modification”.

From UpSetR plot cluster analysis (Figure 4A) we observed that relatively large numbers of unique proteins in each organ likely reflected the inherent features of the individual organs and the diversity of cell types and physiology.

As we observed in previous analysis the largest proteomes were observed in mouse brains (3110) and spleens (3125). Consequently, the two organs exhibited a large number of unique proteins suggesting a distinctive chromatin proteome profile. As expected, due to the chromatin enrichment, we listed the unique epigenetic enzymes relative to brain and spleen. We detected protein associated with transcriptional gene regulation, included histone binding protein of the methyltransferase enzymes and several epigenetic complexes.

8. * "We observed distinctive organ-specific categories, "age-classes and age-development" and categories that reflected their organ source". OK, certainly proteins like "nervous system development" are clearly brain related and "liver development" clearly liver related, so it's nice that those categories are enriched but this isn't really very interesting… it just means the samples are not mislabeled. Regarding the more interesting distinctive "age-classes" and "agedevelopment" that I do not see. What are the "age-classes"? And "age-development" seems vague. True developmental genes will not be especially important at any timepoint the authors' selected, as the first age selected (3 months) is already full adulthood and beyond puberty.

We rephrased this sentence, line 314:

We observed distinctive organ-specific categories (“age-classes and age-development”) and categories that reflected their organ source (Figure 4, Panel C). For instance, unique Go term categories were associated with each organ:

“Nervous system development” and “chemical synaptic transmission” related to the brain;“Cardiac myofibril assembly” and “adult heart development” were attributed to the heart;“Steroid metabolic process” and “liver development” were distinctive to the liver;“Transport” and “sodium ion transport” categories related to the kidney, and;“Angiogenesis” and “respiratory gaseous exchange” were present in the lung.

These results confirmed that many proteins found in individual organs are likely to confer organspecific functions (Figure 4, Panel C). Our strategy showed that a robust enrichment of the chromatin-associated protein in mouse organs was obtained by using GO term analysis.

We reported a significant enrichment of annotated categories listed as age-related, including a large class of molecular features associated with the “core” chromatin environment present in all organs. Our analyses ensured the success of our strategy and confirmed the hypothesis that mammalian organs have ageing-dependent signatures and unique chromatin-associated proteins.

9. * In Figure 5B, the authors find a number of genes that change with age and that are mostly tissue-specific. That's fine, but due to the relatively small sample size and unique tissue signatures, it is hard to be confident in really any of these findings. The authors findings would benefit tremendously with a meta-analysis. For instance, David Glass's lab has a 2019 transcriptional study on liver and kidney gene expression across aging in a conceptually similar study; the hits should be compared. (They also checked hippocampus and skeletal muscle, which may be a worse comparison with brain and heart-but possibly worth checking anyway.) PMID 31533046. This is mRNA and in rats, so much may be different, but when there are overlapping results, that would raise confidence. There are other conceptually-similar studies to compare to as well; off the top of my head there is a 2011 Matthias Mann paper (PMC3033683) that looks at proteome in heart, kidney, and brain. It does not have as many timepoints as yours or the Glass paper, but in other ways is similar.

Glass and Co-authors (PMID 31533046) produced a multi-time point age-related gene expression signature (AGES) from liver, kidney, skeletal muscle, and hippocampus from rats, comparing 6, 9, 12, 18, 21, 24, and 27month old animals. Glass and Co-authors focus on genes that changed in one direction throughout the lifespan of the animal, either early in life (early logistic changes), at middle-age (mid-logistic), late in life (late logistic), or linearly throughout the lifespan of the animal.

The overlap between our proteomic strategy and the transcriptome analysis from Glass and Coauthors may reveal a universal aging signature across mammalians, more specifically across murine. However, there are substantial differences between our strategy and Glass and Coauthors’ approach which may need to be discussed before any meta-analysis is carried out.

Glass and Co-authors used a different murine ageing model in the rat. Rats were chosen because previously studies showed that rats are an excellent model for sarcopenia, which is the age-related loss of skeletal muscle. However, no mention of rat gender was described in the article. Our ageing model was male mouse C57BL/6J. There is extensive literature documenting the moderating effects of healthy aging and gender on cognition in humans (https://www.ncbi.nlm.nih.gov/pmc/articles/PMC3181676/) and aging-related gender differences at a bio-psycho-social level (https://www.karger.com/Article/Pdf/323154). It may be risky to compare data through meta-analysis if the animal models have different genders.

Another critical point is the lack of overlapping between the murine lifespans. Glass and Coauthors harvested organs from 6, 9, 12, 18, 21, 24, and 27 month old animals, which corresponds to middle‐aged, mature adult and old adult mice. We expected a drastic change near where mortality occurs. Our strategy does not completely overlap with the Glass and Coauthors’ rodent lifespan analysis, as only 3 out of the 7 time points are related between the two studies. To exclude any age-related changes due to the alteration of social behavior, including physical characteristics such as motor function and locomotor activity affecting the in vivo dynamics of chromatin changes during ageing, we exclude from our analysis late-stage adult mice in the 18‐24 month old age bracket. Our strategy monitors changes in the chromatinenriched proteomes of mouse organs over a time course that mimics adult development and ageing, from early adult stage upper to middle-aged adult, excluding the stages of lifespan where mortality occurs.

Lastly, Glass and Co-authors’ transcriptome analysis was measured using RNA sequencing technology. The RNA libraries were prepared using the Illumina TruSeq. For this reason, we do not know how large the RNA libraries are, how many gene were included in the custom library, or how many gene readings were performed. The custom gene library may lack of chromatin or nuclear proteins, thus limitation of the meta-analysis may occur during the comparison with our enriched chromatin study.

We found the suggestion from Reviewer #2 extremely interesting. The manuscript certainly contains a good meta-analysis. In the paragraph beginning “Distinct organ ageing profiles are defined by unique protein expression patterns”, line 391, we showed a consistent overlapping between our analysis and a series of human protein biomarker candidates for ageing obtained from the “Human Ageing Genomic Resources” and “GenAge machine learning databank” (supplementary material Figure 5, Panel F) (50) (51). In the brain protein IREB2 is associated with Alzheimer’s disease, whereas the brain protein MAOB is associated with both Alzheimer’s and Parkinson’s diseases. The heart proteins ADD3, PTGIS and COL1A2 are candidates for hypertension and myocardial infarction. The liver proteins INSR, PTPN1 and ENPP1 are associated with diabetes mellitus type 2 and obesity. Lung protein CYP2E1 is related to lung adenocarcinoma and MMP9 is associated with lung neoplasms. In the spleen, the KLK1 protein is a biomarker candidate for hypertension (supplementary material Figure 5, Panel F). In summary, we observed how differentially regulated proteins changed in mice during ageing just like those found in humans.

We inserted the Reviewer #2 suggestion in the discussion, line: 663

10. * Generally, there are a lot of examples picked out for which it is impossible to determine the significance. The authors mention (line 342) that HDAC2 is downregulated in liver and that GATA2AD and TRIM28 are upregulated in kidney. Certainly this could be accurate and mechanistically meaningful, but there are a lot of genes here, we will see a lot of patterns coincidentally. No significance is shown on any of these plots. Based on the fold changes and error bars it looks like it's probably significant, but there's also a ton of multiple testing going on here, which is of course a common issue in exploratory omics analysis. Same concern for the "Reader" enzymes being upregulated in brain and others downregulated in liver. Is this noise? When you do have proteins that are significant in more than one tissue, what's the probability they go in tandem versus opposite? Is it 50/50?

Temporal analysis of the overall dataset, to quantify the protein expressions profile changes during ageing, was performed based on a rank product test. Adjusted q-values were calculated to correct for multiple testing (-log10 qValue < 0.1 cutoff). We used an original approach based on a PolySTest analysis (DOI: https://doi.org/10.1074/mcp.RA119.001777) from Veit Schwämmle and Co-authors were they developed a robust statistical toolbox to improve coverage and depth of large-scale proteomics analysis.

Regarding Figure 5, we observed protein belong to the same chromatin remodeling complex changed expression. For instance, "Reader" enzymes being upregulated in brain and others downregulated in liver. Recently, studies shown the alteration of chromatin complex assembly is largely dynamic. Depending to the cell fate state content the architecture of the chromatin remodelling complex is largely affected. This may explained why "Reader" enzymes being upregulated in brain (Glyr1, Baz1b) and others downregulated in liver (Dpf2, Brd2, Chd2). Also, another reason may the content of the tissue morphology and the diverse “reader” behavior to regulate the gene expression in different cellular contexts.

This result is consistent with the highly dynamic nature of chromatin remodelling complexes. Some protein subunits are only present in a complex at distinct time-points to provide a unique function or feature.

We address the Reviewer #2 suggestion in the manuscript line: 380. We also suggested relevant literature about the alteration of chromatin remodelling assembly

References

Dynamic Protein Interactions of the Polycomb Repressive Complex 2 during Differentiation of Pluripotent Cells: doi: 10.1074/mcp.M116.062240

Dynamic Competition of Polycomb and Trithorax in Transcriptional Programming doi: https://doi.org/10.1146/annurev-biochem-120219-103641

A Family of Vertebrate-Specific Polycombs Encoded by the LCOR/LCORL Genes Balance PRC2 Subtype Activities doi: https://doi.org/10.1016/j.molcel.2018.03.005

11. ** The shape of the volcano plot in Supplemental Figure 5D – an almost perfect U – indicates a rather aggressive normalization procedure that almost ties fold change to p-value. This is maybe unavoidable, but it's not ideal. The log2 fold changes are also very large – log2FCs of -5 and +7? So some proteins are upregulated 128-fold in the spleen across age? In general the log2FCs here are far larger than I've seen in any study. For instance, the Mann paper above (PMC3033683) finds only 1% of proteins that change by more than 2-fold between 5 and 26 months. My own studies (although not on aging) also find far smaller fold changes. I don't think I've ever seen a fold change of larger than 2^3^ unless the starting value was so low as to be essentially noise. That would be also my concern here; are the FCs that are like 2^8^ changing from say, 0.01 to 1.0? The aggressive normalization would do this kind of distortion. In my experience normally the very large FCs also have relatively low p-values for that exact reason.

While mouse strains should provide the lowest variance out of all appropriate model animals, often the variance between them is not as low as expected. Also, to avoid any strong bias towards both mouse organ and ageing, each organ has been normalised independently (supplementary figure S4, boxplot analysis)

Due to high complexity of the in vivo model we applied a powerful statistical tool using an original approach based on a PolySTest analysis (DOI: https://doi.org/10.1074/mcp.RA119.001777). PolySTest is a robust statistical toolbox to improve coverage and depth of large-scale proteomics analysis based on large numbers of different variables.

Temporal analysis of the overall dataset able to quantify the protein expressions profile changes during ageing was performed based on a rank product test. Adjusted q-values were calculated to correct for multiple testing (-log10 qValue < 0.1 cutoff).

The proteome dataset was obtained from organ samples harvested during adult mice lifespans. We focused on proteins that changed in one direction during the ageing of the organ, or linearly throughout the lifespan of the animal. We applied a novel statistical strategy to detect the most significant chromatin proteome changes associated with ageing. We used rank product tests to identify proteins that exhibited significant abundance changes during ageing (supplementary material, Figure 5, Panel D). We called these “differentially regulated proteins”.

We attached on supplementary material Figure S4 the brain chromatin enrichment undergo PolySTest analysis. We observed how the PolySTest analysis help our data interpretation and provide a robust strategy to identified significant and differentially regulated proteins over time.

12. * Are the tissues normalized together, or each separately? It does not seem like the values are ever directly compared, in which case normalizing separately would probably be better due to the relative paucity of common proteins across all six tissues, but I could see an argument made for either case. In the volcano plot for liver in Supplemental Figure 5D, why is the volcano not centered on FC=0? It is shifted dramatically to the right; this looks like a mistake in the drawing as the significance for minus log2FC starts at like FC=2^1^ (red), but for positive log2FC it is only significant at like FC=2^4^ which doesn't make sense for the U shape.

The proteome dataset was obtained from organ samples harvested during adult mice lifespans. We focused on proteins that changed in one direction during the ageing of the organ, or linearly throughout the lifespan of the animal. A statistical strategy was applied to detect the most significant chromatin proteome changes associated with ageing.

First, to avoid any strong bias towards both mouse organ and ageing, each organ has been normalised separately (supplementary figure S4, boxplot analysis)

Subsequently, we used rank product tests to identify proteins that exhibited significant abundance changes during ageing (supplementary material, Figure 5, Panel D). We called these “differentially regulated proteins”.

Depending on the experiment’s intrinsic data structures and variations, rank product tests indicated where performance was required to improve in a complementary manner (DOI: https://doi.org/10.1074/mcp.RA119.001777). An example of this is where we showed a comparison of protein expression profiles in a mouse brain aged from 3 to 15 months using different statistical tests. By maintaining the same threshold cut-off across all tests, a different “V” Volcano shape was observed. We hypothesized the alteration of the canonical “V” shape due to:

a)Large differences in the protein expression profile from young adult mice (3 months) in comparison to “upper middle-aged” mice (15 months), possibly due to the label-free strategy;

b)The intrinsic biological differences present in the organ; for example, the large and different morphology and cell type compositions.

To avoid any strong bias towards both mouse organ and ageing, the rank product test method was selected. In common with Rainer Breitling and Co-author (DOI: 10.1016/j.febslet.2004.07.055), we observed that the rank product test performed well for our proteome strategy and it identified significant biological feature changes during time. Subsequently, we performed the same statistical analysis for the rest of the organs. We recommend the use of the PolySTest tool for future proteome analysis with a large number of variables. PolySTest tool is a free statistical analysis platform available on:

http://computproteomics.bmb.sdu.dk:8192/app_direct/PolySTest/

13. * Near line 381, the authors mention that" the highest unique changes were observed in brain, heart and kidney organs relative to.… "development/ageing"", but not for lung, liver, and spleen (I thought the authors said they removed spleen from future analyses in line 330 so I am surprised to see it in Figure 6, but I digress). What is this annotation of "gene expression" "ageing/development" etc from? Is it the authors or from GO? I have not seen it clustered like that in DAVID, but maybe it is a new feature. In any case, the categorization seems somewhat strange to me. For instance, "cellular respiration" is listed as "ageing / development" – but cellular respiration is involved in basically everything from "cellular metabolic process" to "gene expression" and more. This is even more confusing when "ATP metabolic process" is listed as "cellular metabolic process" but "cellular respiration" is listed as "ageing/development". Why would "ATP metabolic process" be separate from "cellular respiration"? I know these are not exactly the same, but they are so similar I would certainly cluster them together in terms of basic ontological function. Why is "heart contraction" listed as an "ageing/development" function? Some processes in "ageing/development" make sense (e.g. "double-strand break repair") but the four broad ontologies seem largely arbitrary.

We rephrased the sentence, Line 415:

“We reported a high proportion of uniquely annotated categories relative to each organ. For instance, the highest unique changes observed in the brain were relative to “gene expression”, “ageing/development”, and “structure organization and biogenesis”; in the heart and kidney significant changes were observed relative to “structure organization and biogenesis”; the liver showed changes across the “gene expression” and “structure organization and biogenesis” categories; the lung showed the highest unique changes in the “ageing/development” category, and; relative changes in the spleen were detected at the “gene expression” level”

Reviewer #2 pointed out that the four broad ontology groups in Figure 6 are largely arbitrary. In the paragraph “Functional analysis of ageing organ chromatin-enriched proteomes***”***, we applied an extensive Gene Ontology (GO) analysis to characterize the overall common and unique pathways differentially regulated during ageing across all the organs.

From Figure 5B we first extrapolated the “unique differentially regulated proteins” in each organ. Subsequently, we applied Gene Ontology (GO) analysis across all the “unique differentially regulated proteins” to list the overall common trends and processes being differentially regulated during ageing across all the organs. We sorted the annotated categories by their relative GO term category and separated them by their main family source.

Several pathways were observed to change the chromatin conformation induced by stress responses affected by ageing. We also detected categories associated with modification of the cell structure environment due to organ development. By listing each single category, we described a comprehensive biological profile and listed the main common pathways affected by age-responses present in all organs and related to differentially expressed proteins (Figure 6).

In line with the previous GO analysis (Figure 4B and 4C) we observed the most significant categories changed in chromatin conformation and cell development induced by stress responses affected by ageing. Therefore, we largely sorted four constitutive parts and labeled them as “Gene Expression”, “Ageing / Development”, “Cellular metabolic process”, and “Structure organization and biogenesis” to create a comprehensive cell biology profile during ageing for all organs. Subsequently, the differentially regulated proteins were sorted by their organ source and subjected to further GO analysis in order to distinguish unique organ pathway responses as distinct to those associated with ageing (Figure 6).

Overall, using GO term analysis we dissected the biology landscape aging-related through adult mouse lifespan. By breaking down common and unique regulated categories we aimed to address new biological avenues not yet assigned to be known as ageing-related pathways and improve gene-annotation enrichment analyses related to chromatin cell environment

15. ** The western blots in Figure 7 are a reasonable approach towards such validation, but without the raw data, I don't know what is exactly expected for each of these genes (although the green and red colors at top give me a clue, they are not precise). Is it expected that RNF20 goes down in the aging brain? This gene is not among those shown in Figure 5C. The upregulation of proteasome genes in kidney is surprising; proteasome normally declines with age. Why were these proteins selected? Did you select other proteins that you did not show because they did not validate? Are these measurements in the exact same tissues as the proteomics? That would be an OK validation but essentially just show that the digestion and mass spectrometry worked, rather than independently showing that these proteins change with age – rather like doing a qPCR to validate an mRNA change in the same samples that you ran RNA-seq. This is nice enough to do and appreciated, but it validates the technology and data measurement rather than the actual biology. Also how many gels am I looking at here? I see bactin and RING1b both at the same location in the brain, so where is the loading control for RING1b? I assume this was correctly done as the westerns look well done, but the authors should show the full gel and/or exact loading control. Similar concerns with all of the other tissues, e.g. SIRT3 and SIRT5 are the same molecular weight, so what was the loading control for those 8 lanes?

Following Reviewer #2’s Note (15), we have taken the comments on board to improve and clarify the manuscript. Figure 7 attempted to confirm independently, by a semi-quantitative approach (western blotting), the observations noted in our wider dataset, specifically the unique co-expression network generated. We looked in more detail at the most significant organspecific up-regulated and down-regulated biological processes during mouse ageing.

To avoid cherry-picking in the validation, we divided our strategy into two parts:

Part A: We used a STRING database to investigate the protein/protein interaction (PPI) of chromatin associated proteins belonging to the most significant regulated organ-specific pathways (up-regulated and down-regulated) changes during mouse ageing;Part B: Integrating protein/protein interaction data with the quantitative protein expression profile.

By integrating protein/protein interactions and protein expression we derived co-interaction and co-expression networks to improve our understanding of the biological mechanisms involved in ageing. We attempted to confirm independently, by immunoblotting, the observations noted in our wider dataset, specifically the co-expression network (Figure 7, Panel B). We then attempted to validate the most significant organ-specific biological processes that are regulated during mouse ageing, particularly at the chromatin level, as we hypothesized that changes in protein expression affects each organ in a distinct manner.

We validated the protein up-regulation changes in the brain, heart, and kidney and conversely the down-regulation changes in the liver and kidney. As we explained in the manuscript, the spleen was not validated as only a few differentially expressed proteins were observed there, and they did not allow for useful GO and protein network analysis. In Figure 7B the biochemical validation of four proteins module responses to ageing were analyzed. Chromatin lysates (from the brain, heart, liver, and kidney) were immunoblotted with the indicated antibodies. Loading controls were used for the proper interpretation of the western blot results. Β-actin and Histone H3 confirmed that protein loading was the same across the samples during ageing.

To compare the trend change (up-regulated and down-regulated) in all four-protein modules further control validations were applied. We probed proteins which exhibited either smaller expression changes or opposite trends. For example, in the brain module, the RNF20 antibody was probe for antagonists in the up-regulation trend. RIG1B showed only minor protein expression changes. The same loading strategy was used with the rest of the organ modules. A useful tool for the analysis of all four protein modules was the integration of quantitative data: a bar above the blots corresponding to the quantitative protein expression levels determined in our proteomics experiments over time (green for up-regulated and red for down-regulated).

Reviewer 3

1. Regarding the preparation methods, I think it would be more accurate to label the proteome samples as "nuclear proteome" instead of "chromatin-enriched proteome".

We agree that the discrimination between nuclear and chromatin compartments is relatively small. This is in part due to the lack of protein databank cell localization data for the chromatin compartment. We demonstrate that our protein isolation protocol successfully enriches for chromatin. In Figure 1A, we demonstrated that the polycomb protein SUZ12 was detected in the chromatin enriched fraction but not in the nuclear fraction. In chromatin fractions isolated from mESC cells we demonstrated that many known chromatin-associated protein complexes (such as Polycomb Repressive Complex 2 (PRC2), Nucleosome Remodeling and Deacetylase (NuRD), BRAF-HDAC complex (BHC), and mixed-lineage leukemia (MLL) complex) were detected by mass spectrometry (Figure 1B). In the mouse brain we showed, using this protocol, that the chromatin marker histone H3 was present in the chromatin enriched fractions while the cytosolic marker GAPDH was reduced (Supplementary material, S2). We determined that these fractions, while they contained many nuclear components, are enriched chromatin-associated proteins in comparison to total brain lysate. The dynamic range plot was used to visualize the mESC chromatin fraction which was highly enriched for chromatin-associated proteins as compared to the whole cell lysate (Figure 1, panels B and C).

In conclusion we demonstrated that the chromatin enrichment protocol used for mESCs is applicable to mouse organs as shown in Supplementary Material Figure 2.

References

Sanders et al. 1978 (Sanders 1978)- original salt extraction protocol

Henikoff et al. 2009 (Henikoff et al. 2009)- updates the Sanders protocol and describes isolating different types of chromatin (and the associated proteins, but the focus is on the chromatin itself).

Teves & Henikoff 2012 (Teves and Henikoff 2012)- details the protocol used in their 2009 paper.

Herrmann et al. 2017 (Herrmann et al. 2017)- details a protocol focused on the isolation of chromatin-associated proteins from total chromatin (as opposed to isolating different types of chromatin).

2. Line 183: 15 months of age is not an old mouse and the time points do not actually probe the lifespan of a mouse. However, the selection of the time points is rational, and also justified in the changes that can be observed. It might be good to point out that the time point of 15 months is in mid-life before mortality begins to diminish a population.

We thank the reviewer for this helpful comment. We agree that 15 months represents the upper middle age rather than old age in mice. As the reviewer mentioned, after this time point mortality begins to diminish in the population and it is challenging to harvest enough mice for analysis after this time point. However, we do observe distinct ageing profiles across organs for the time points examined in our study which suggests that this time point is sufficient to observe age related changes. We revised the text in the section “Quantitative chromatin proteomics of ageing mouse organs” to make this more clear.

3. The nuclear enrichment ranges from 30% to 65% across tissues. It would be important to discuss how this difference might affect comparisons.

We thank the reviewer for sharing this observation. Differences in nuclear and chromatin enrichment across tissues are an important consideration. We added a sentence in the section “Quantitative chromatin proteomics of ageing mouse organs”, line 208 to discuss how the difference in enrichment ranges might affect the organ comparison.

4. In Lines 238 and below: for a better understanding of the experiment […]

We thank the reviewer for their helpful suggestion. We changed the section title to “Quantitative proteomics define biological changes in the ageing process” for a better overall meaning of the paragraph.

5. Table 3 it is not clear

We revised the text to convey the concept: line 253

To define ageing signatures across mouse lifespan in each organ, we performed a comprehensive functional analysis of the proteomics datasets using an UpSetR plot to investigate ageing markers.

To gather more information from the UpSetR plot, we retrieve the dataset to estimate the amount of protein shared or not in all mouse organs over time through to their cell compartments. In line with our results Spleen and Brain showed the highest number of unique chromatin associated protein. We also estimated a large number of proteins found in individual organs are likely to confer organ-specific functions.

6. With the collection of nuclear extracts from the various organs, the study results might be confounded by tissue heterogeneity and temporal changes in organ composition. At baseline, what are potential contributions from the various cell type in the organs? Considering the many cell types contributing, for example, to brain extracts, how should the results be interpreted for neurons, glia, etc.? What is the contribution of blood and blood vessels and immune cells? Finally, their relative contribution is likely to change with age, as for example inflammation increases with time. It would be great if the authors could discuss this in the manuscript.

We agree that our results observed in bulk tissues might not be reflective of the alteration of the chromatin landscape at the singular cell level. We address this in line 696.

We are aware that the nuclear and chromatin proteome analysis from the various organs may not be reflective of the overall cellular heterogeneity compositions of the tissues. Chromatin enrichment was, however, observed among each tissue demonstrating the high reproducibly and reliability of the experimental approach, with a coefficient variation estimated at less than 10% across all the samples.